# Superelastic Tellurium Thermoelectric Coatings for Advanced Trimodal Microsensing

Shaowei Cui [1,7], Linlin Li[2,7], Zi-Xin Huang [3,7], Yanzhe Yu[1], Mingxue Cai [4], Xiangyin Bao[1], Chaofan Zhang[1], Tiandong Zhang [1], Long Cheng[1], Wenxuan Zhang[2], Zheng Lou [2], Shuo Wang[1], Wen Gong[5], Chao-Feng Wu [6] ✉, Lili Wang [2] ✉ & Yu Wang [1] ✉

Tactile endoscopes can provide physicians with rich sensory information, enabling fast and accurate medical diagnoses. However, existing tactile endoscopy sensors do not consider temperature perception, which is a very important diagnostic indicator in medicine. Here, we report for the first time a tellurium-based superelastic thermoelectric visual-tactile sensor. This platform achieves a breakthrough by combining tellurium thermocouples designed based on crystal structures with viscoelastic silicone encapsulation, enabling simultaneous microscale visual, thermal, and force measurements in a single device. By employing a morphologically optimized tellurium-polymer heterointerface and advanced deep neural network algorithms, we address the inherent trade-off between transparency and responsiveness, achieving artifact-free imaging, real-time thermal mapping, and microstructure force feedback. We conduct clinical endoscopic palpation experiments on live rabbits and successfully achieve tactile diagnosis of inflamed tissue including temperature distribution, especially in cases where visual distinction is difficult, pointing out possible development directions for intelligent endoscopy systems.

Tactile endoscopes can significantly enhance the richness of sensory information in tactile diagnosis and minimally invasive surgery, thereby improving doctors' efficiency and accuracy[1,2]. In the past decades, tactile endoscope systems have achieved force measurement while maintaining basic visual observation capabilities[3–6]. This marks progress in enhancing the safety and accuracy of endoscopic procedures by enabling surgeons to obtain both visual insights into the surgical site and quantitative force feedback[2,7,8]. However, a notable limitation of current tactile endoscope systems lies in their general lack of temperature-sensing functionality, which is a critical capability for endoscopic surgeries. The involvement of temperature modalities can achieve necessary thermal monitoring[9,10] (real-time target tissue temperature measurement to avoid thermal damage) and play a key role in the accurate diagnosis of abnormal tissues[11–13] (such as identifying inflammation and tumor lesions through local temperature changes). Therefore, how to seamlessly integrate temperature sensing

[1]State Key Laboratory of Multimodal Artificial Intelligence Systems, Institute of Automation, Chinese Academy of Sciences, Beijing 100190, China. [2]State Key Laboratory of Semiconductor Physics and Chip Technologies, Institute of Semiconductors, Chinese Academy of Sciences, Beijing 100083, China. [3]School of Electrical and Information Engineering, Wuhan Institute of Technology, Wuhan 430205, China. [4]Shenzhen Institute of Advanced Technology, Chinese Academy of Sciences, Shenzhen 518000, China. [5]Research Center for Advanced Functional Ceramics, Wuzhen Laboratory, Jiaxing, Zhenjiang 314500, China. [6]Center of Advanced Ceramic Materials and Devices, Yangtze Delta Region Institute of Tsinghua University, Zhejiang 314006, China. [7]These authors contributed equally: Shaowei Cui, Linlin Li, Zi-Xin Huang. ✉e-mail: wuchaofeng@tsinghua-zj.edu.cn; liliwang@semi.ac.cn; yu.wang@ia.ac.cn

technology into tactile endoscopy systems to achieve high-precision 3D force measurement and high-definition visual imaging without compromising the system's existing compact structure remains a significant challenge in this field.

Among the mainstream temperature measurement methods, non-contact temperature sensing represented by infrared temperature sensors is favored because it avoids contact damage, but the measurement accuracy is limited[14–16]. In contrast, contact-based temperature sensors are relatively mature in development and application, primarily categorized into thermistors and thermocouples. Thermistor made from precious metals like Pt and Cu offer exceptional signal stability[17–19]. However, their minimal signal variation and complex calibration circuits make them difficult to integrate onto flexible micro-probes for visual-tactile sensing, rendering them more suitable for temperature drift calibration in micro-nano electronic devices[20–23]. Thermocouples based on the Seebeck effect show a strong correlation between their signal and contact position and have excellent spatial resolution[24–26]. Leveraging the high band degeneracy and unique lattice arrangements of tellurium materials, a series of outstanding thermoelectric materials have been developed around tellurium and its compounds[27–31]. However, due to thermoelectric mechanisms, thermocouple systems represented by tellurium-based compounds typically require larger longitudinal or lateral dimensions to achieve greater temperature differentials. This inevitably impacts signal acquisition and reconstruction in tactile sensing. Therefore, the construction of 3-dimensional, small-size, high-sensitivity temperature sensor devices remains an extremely significant challenge.

Here, we develop a paradigm-shifting Te endoscope architecture (T-scope) that achieves unprecedented tri-functional integration of vision, 3D tactile force sensing and sensitive thermal analysis within an ultra-compact 4.8 mm probe. To prevent the temperature sensor from interfering with the visual and tactile information, we employ the high Seebeck coefficient and weak thermal conductivity of Te polycrystalline thin film to construct a 3-dimensional, small-size temperature sensor on the surface of silica gel. The thickness of the temperature-sensing film is approximately 200 nm, with a lateral size of less than 1 mm² and a detection accuracy of 647 μV/K. The AI-driven visual-tactile deep neural network enables accurate three-dimensional contact force estimation for tissues with different hardness. It also integrates visual restoration observation integrated with spatiotemporal context information, by performing pixel-level segmentation and tracking of the artificially marked deformations naturally provided by Te devices[32–34]. The real tactile endoscopy experiments conducted in living rabbits verify its leading clinical advantages: it successfully distinguishes abnormal inflammatory tissue from normal tissue from aspects such as visual observation, pressure mechanical properties and temperature distribution.

## Results

### Principles and design of the T-scope sensor

The core innovation of our study lies in integrating cutting-edge flexible thermoelectric materials with visual AI technology to realize a visual-tactile-thermal tri-modal sensing system within compact endoscopic devices (Fig. 1a, Supplementary Movie S1). Our first contribution is the design of a tactile probe that can seamlessly integrate into existing endoscopic systems. The probe is made of a transparent elastomer with specially designed textured patterns, as illustrated in Fig. 1b. This design enables the camera to detect tiny deformations in the sensor caused by physical contact, achieving temperature and force modeling while maintaining clear visual observation (Fig. 1c).

The probe is integrated at the front end of a traditional endoscope. In this study, an over-molding process is employed to construct a soft-rigid hybrid structure. This structure consists of two components: a flexible elastomer (Supplementary Fig. S1) for high sensitivity contacts detection and a rigid skeleton for structural support. The transparent silicone substrate of the probe is precisely molded onto the prefabricated skeleton using advanced casting technology, with texture layers and protective layers subsequently applied. The final sensor combines the advantages of both components—it maintains structural stability to ensure sufficient strength under large contact forces, while exhibiting high sensitivity in detecting and responding to minimal interaction forces.

A critical breakthrough involves using tellurium polycrystalline films based on the Seebeck effect to embed temperature-sensing functionality into textured patterns. This flexible material seamlessly integrates with elastomers, enabling precise temperature measurement while preserving tactile imprint detection and image quality (Fig. 1d). The T-scope system translates tactile information into identifiable pattern variations through regularly arranged tellurium polycrystalline patterns on its elastomer surface. The process of transferring its 3D microstructure enables adhesion to micro-level curved surfaces, with each thermoelectric imprint maintaining 200 μm diameter. To ensure durability in complex biofluid environments, a protective layer is added via secondary injection molding (detailed manufacturing steps in Methods section, Supplementary Fig. S2).

In addition, we propose advanced deep neural network methods for visual observation inpainting and accurate 3D force estimation (Fig. 1e). Specifically, we propose the EndoForce network, which performs 3D force estimation based on the thermoelectric imprint changes in various contact states, and a transformer-based video inpainting algorithm, which provides imaging quality comparable to endoscopic systems without a tactile layer and high-sensitivity.

### Structure of Te-based thermoelectric subsystem

The thermoelectric component based on polycrystalline tellurium thin films is developed for temperature sensing in the T-scope system (Fig. 2a). In the subsystem design, highly transparent ITO electrodes are employed as external leads to effectively reduce optical interference with the T-scope vision system. By precisely controlling the total device thickness at the 10 μm level, interference effects on the deformation processes of posterior organic materials are significantly minimized. The innovative selection of polycrystalline tellurium as the sensitive functional layer features a crystalline structure composed of parallel-aligned ternary helical chains (Fig. 2b). Intra-chain Te atoms are bonded through intermolecular interactions, while inter-chain ordering is maintained using van der Waals force-driven stacking. This unique crystal structure exhibits significant lattice scattering, effectively suppressing lattice heat conduction and promoting the establishment of a significant temperature gradient (Fig. 2c). This enables efficient thermoelectric conversion and reduces the trade-off between thickness and responsivity.

The fabrication process flow comprises the following key steps (Fig. 2d, Further details in the Methods section and Supplementary Fig. S3): Initial surface cleaning and PDMS composite treatment of 10 μm PET substrates; Magnetron sputtering deposition of gold electrodes on half-area substrates followed by full-substrate tellurium thin film deposition; Laser direct writing processing of AuTe-Te interfaces into 1 mm-diameter temperature-sensitive unit (Fig. 2e). When placed on a 42 °C hot plate, the temperature-sensitive unit demonstrates clear temperature field distribution characteristics on its surfaces, confirming temperature detection capabilities. Furthermore, an ITO polycrystalline thin film is fabricated and patterned as an external electrode. Optical photography reveals the excellent light transmittance of the ITO layer (Fig. 2f). Finally, the temperature-sensitive unit is integrated with the ITO electrode through microassembly, completing the fabrication of a Te-based polycrystalline temperature sensor (Fig. 2d).

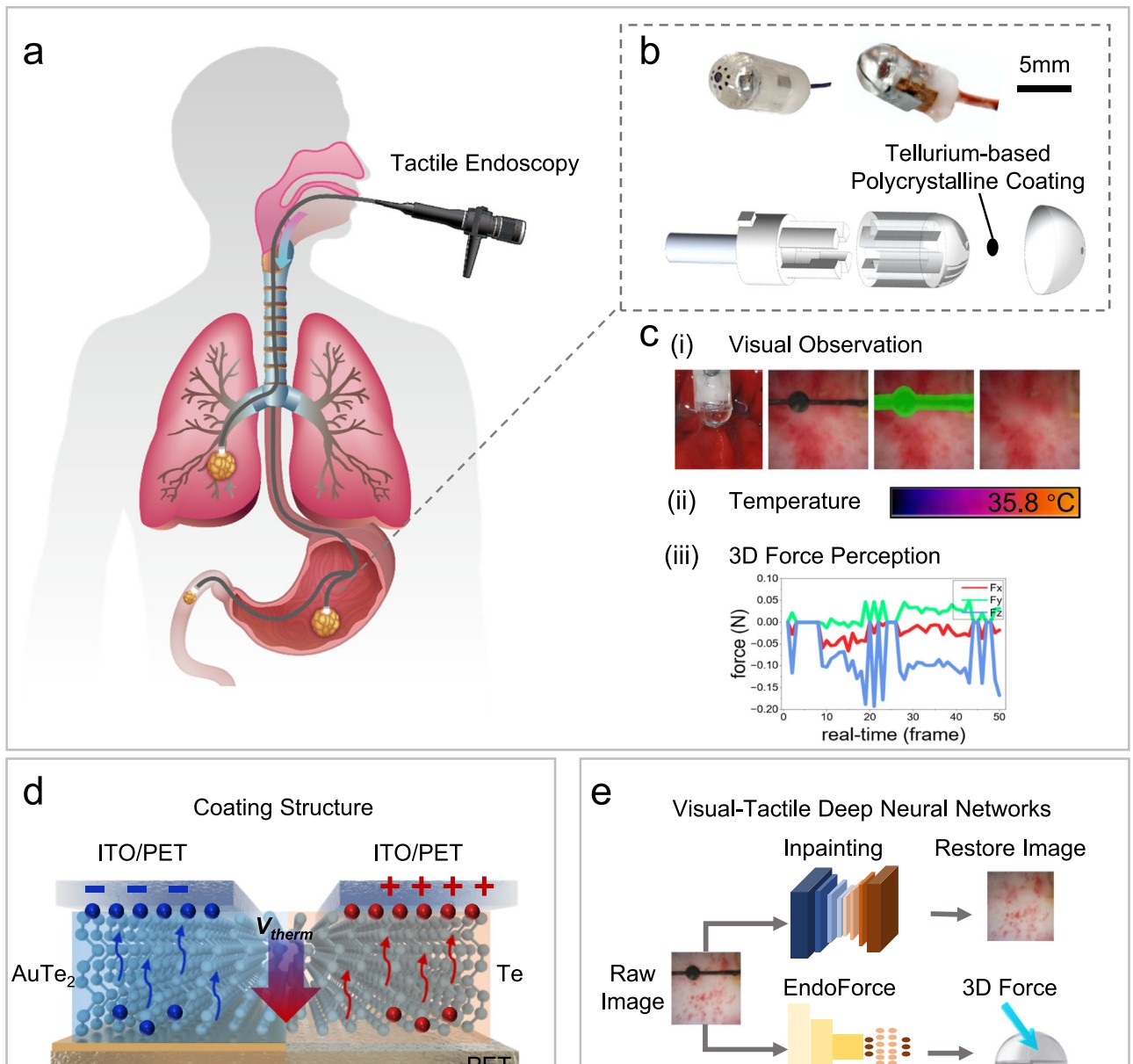

**Fig. 1 | Architecture and functional integration of the T-scope sensing system.**
**a** The compact sensing module (5 mm cross-section) demonstrates endoscopic compatibility through front-end integration, facilitating pathological tissue identification in visceral lumens including pulmonary bronchi, gastric compartments, duodenal regions, and other mucosal surfaces within aerodigestive pathways. **b** Disassembled perspective illustrating component hierarchy and mechanical assembly of the T-scope. **ci** The restored visual observation provided by the visual inpainting neural network. **cii** Temperature sensing provided by the Te-based thermocouple coating subsystem. **ciii** The perceived 3D force sequence provided by the tactile neural network. **d** Thermoelectric transduction mechanism with layered material configuration (ITO Indium Tin Oxide, PET Polyethylene terephthalate, $V_{therm}$ Thermoelectric potential (V)). **e** Contact 3D force mapping and endoscopic image restoration from raw images based on visual-tactile deep neural networks. Source data are provided as a Source Data file.

## Simulation and Evaluation of Te-based Thermoelectric Performance

For assessing the device's operational viability, finite element analysis (FEM) through COMSOL Multiphysics is implemented to characterize thermal-to-electrical conversion performance in simulated application scenarios (Fig. 2g). The model incorporates a PET substrate to mimic surface encapsulation and an Ecoflex layer to replicate integration with visual sensors, while considering natural convection effects. By applying thermal gradients (20-40 °C) at contact interfaces, simulations reveal significant temperature field redistribution and corresponding thermal potential variations (Fig. 2h, Supplementary Fig. S4). At a contact temperature of 30 °C, a 4 °C temperature gradient across

the Te domain produces a 4 mV output, demonstrating its robust thermal detection capabilities (Fig. 2i). The difference in thermal behavior between the Te and AuTe alloys is evident: AuTe maintains a uniform temperature distribution due to its high thermal and electrical conductivity, while Te's lower lattice thermal conductivity results in a 10 °C temperature gradient at a 40 °C contact temperature. This contrast amplifies carrier accumulation on the Te surface, generating a 9 mV potential difference (Fig. 2h), confirming the mechanism of temperature-dependent signal generation.

Subsequently, the thermoelectric performance is evaluated under controlled contact heat sources (20–42 °C). The Te-based thermoelectric sensor exhibits stable output potential at identical contact

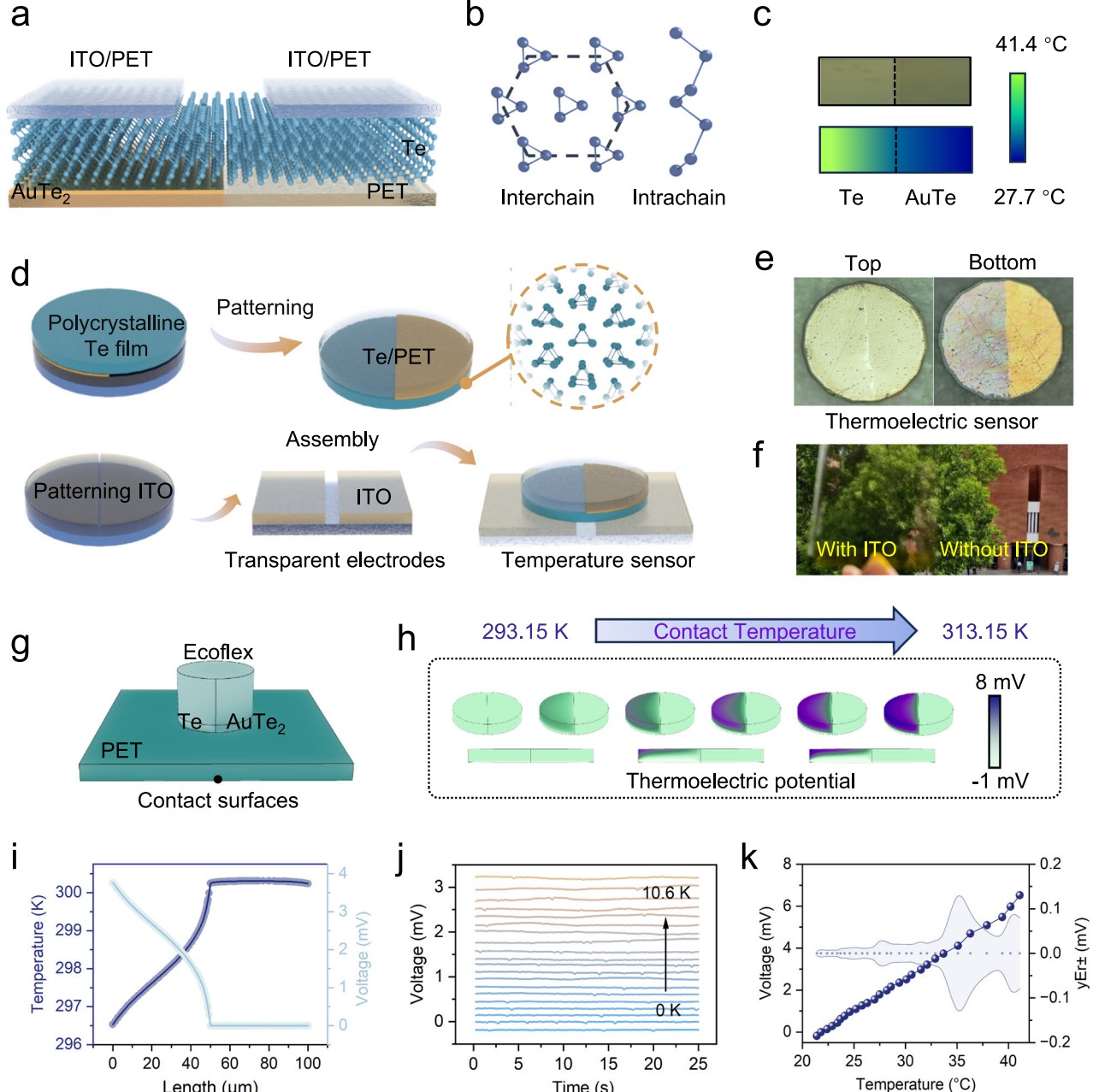

**Fig. 2 | Mechanism and evaluation of the thermoelectric subsystem. a** The structure of the Te-based thermoelectric subsystem. **b** The lattice structure diagram of Te. **c** Thermal gradient under 42 °C thermal loading (Upper: Macroscopic observation; Lower: Infrared thermographic mapping). **d** Fabrication process flow for polycrystalline Te temperature detector (Magnetron sputtering thin-film deposition combined with laser-assisted micro-patterning). **e** A shot of the finished Te-based thermoelectric sensor. **f** Optical image of magnetron-sputtered indium tin oxide (ITO) films deposited on flexible polyethylene terephthalate (PET) substrates, demonstrating high optical clarity and visible-light transmittance. **g** Computational simulation illustrating the working principle of thermoelectric sensing elements, incorporating an elastomeric contact layer at the top (to simulate the thermoelectric subsystem immobilized on the tactile interface) and a thermally regulated boundary at the bottom (to simulate different contact temperatures), with colormap visualization of the thermal distribution. The figure utilizes a contact temperature of 30 °C and an ambient temperature of 25 °C. **h** The Seabeck-induced voltage mapping within the functional layers, quantitatively correlated with interfacial temperature modulation at the active contact region, 20-40 °C. **i** Distribution of thermal and electrical potentials under 30 °C contact temperature along the upper end of the subsystem. **j** Time-dependent voltage response characteristics of the device under varying thermal differentials, defined as the difference between the contact temperature and the ambient temperature. **k** Quantitative dependence of generated electrical potential on applied temperature. Source data are provided as a Source Data file.

temperatures, demonstrating its stability (Fig. 2j). Further analysis reveals a linear relationship between voltage and contact temperature variation, with a sensitivity of 0.323 mV/K (Supplementary Fig. S5). Remarkably, when heated to 39 °C (close to mammalian body temperature), the output is 5.5 mV (Fig. 2k). Measurements reveal a temperature difference between the upper and lower surfaces, and

analysis reveals the film's Seebeck coefficient to be 0.647 mV/K, 2.6 times higher than that of bulk tellurium (Supplementary Fig. S6). These results highlight the dual advantages of this sensor: Te's low thermal conductivity facilitates the formation of steep thermal gradients, while its enhanced Seebeck response ensures high-resolution temperature detection, confirming its suitability for environmental monitoring and

in-vivo biomedical applications. Detailed thermoelectric effect analysis can be found in Supplementary Note S1.

## Visual-tactile deep neural networks

Because the proposed thermoelectric devices provide natural landmarks in the endoscopic field of view, our endoscopic vision system can capture their position and overall shape changes, thereby creating an accurate map of the 3D contact forces acting on the probe[35,36]. We first design a tactile data acquisition system (Fig. 3a(i)): two T-scope tactile endoscopic probes with different marker patterns (Fig. 3a(ii)) are fixed on a commercial multi-dimensional force sensor. Contact tests with varying directions and magnitudes are conducted using a variety of soft materials that fully cover the hardness range of human tissues (elastic modulus from 0 to 100 kPa) (Fig. 3a(iii), Supplementary Fig. S7). The six-axis force/torque sensor provides ground-truth 3D forces acting on the surface of the tactile probe. Subsequently, we adopt a data-driven approach and propose the EndoForce 3D force estimation network to directly estimate 3D forces from raw visual observation inputs.

The proposed EndoForce network (Fig. 3b) first extracts thermoelectric imprint information through an image segmentation network (Efficient-SAM[37]). By calculating the difference between real-time imprint segmentation results and non-contact reference values, positional displacement differences are isolated. Binary XOR processing highlights the movement of markers, and the processed images are then fed into the force estimation network. This network is based on the ResNet architecture, with detailed information provided in Supplementary Fig. S8, Note S2, and Table S1. This method effectively reduces interference from diverse endoscopic backgrounds, allowing the network to focus on learning pattern features at the thermal-electrical imprint level (Supplementary Table S2).

Experimental results demonstrate that the system achieves excellent performance under varying contact magnitudes and directions (Supplementary Table S3 and S4). The average full-scale error percentage of the contact force is about 6%FS, with the average errors of the tangential force and normal force components being 0.02 N (6.50%FS) and 0.06 N (5.96%FS), respectively. Error distribution analysis reveals consistent prediction accuracy across different force magnitudes, with no error escalation observed when contact forces increase (Fig. 3c(i)). Directional error analysis (Supplementary Fig. S9) indicates comparable performance across different orientations, though slightly higher errors occur in the central and downward directions. This phenomenon may be attributed to the contact habits of operators. Further validation through randomized finger contact trials (Fig. 3c(ii)) shows stable 3D force estimation with consistently low errors, confirming the generalization capability of the network. The testing process of perceived 3D force versus ground-truth 3D force when randomly pressing the sensor probe with a human finger is available in Supplementary Movie S2.

Thermoelectrical imprints cause visual occlusion in endoscopic images (Fig. 3a(ii)), which may compromise the clarity of the diagnostic field of view. To address this issue, we adopt ProPainter[38] for AI-driven visual inpainting. We first prepare a dataset suitable for real in-vivo tissue scenarios (Fig. 3d(i), Supplementary Note S3). Specifically, we use a T-scope probe without thermal-electrical markers to capture videos of the respiratory tract and gastric tissues of live rabbits. Visual inpainting synthetic datasets are then generated by manually adding various markers, including different dot arrays and dot-line patterns (Supplementary Fig. S10).

We first retrain the ProPainter network using the synthetic dataset with ground-truth labels. In real in-vivo experiments, the visual inpainting network with the trained weights is directly used for real-time visual observation (Fig. 3d(ii)). Results from the in-vivo experiments show that the cross-domain trained model successfully reconstructs visual information occluded by different thermoelectrical imprints (Fig. 3e(ii)). More inpainting examples can be found in Supplementary Fig. S11 and S12. The model achieves an average peak signal-to-noise ratio (PSNR) of 37.35 decibels (dB) across various contact scenarios. Meanwhile, it maintains a complete diagnostic field of view and remains unaffected by variations in the position and size of marker occlusions (Fig. 3e(i), (ii), Supplementary Table S5).

## In vitro simulated visual-tactile diagnostic experiment

To verify the practical efficacy of the T-scope system in clinical endoscopic environments, we conduct visual-tactile diagnostic experiments using full-scale biosimulation phantoms. During the experiments, the T-scope sensing device is mounted at the distal end of a standard endoscopic system. Operators perform multiple contact detection tests by manually manipulating the endoscope within the simulated models of the human bronchus, stomach, and large intestine (Fig. 4a(i), Fig. 4b(i), Fig. 4c(i)). This study specifically records the detection data from the sensor's interaction with physiological and pathological tissues in these anatomical structures, and conducts comparative analyses on the dual-modal sensing capabilities (three-dimensional contact force and vision) of the T-scope system when it contacts normal tissues and simulated tumor tissues.

Biomechanical test data from the bronchial and gastric models (Fig. 4a(ii), b(ii)) shows that due to the higher biomechanical stiffness of tumor tissue, the fluctuation of its normal force value is more significant than that of healthy tissue when maintaining a stable contact state. Visual inpainting results (Fig. 4a(iii), b(iii)) indicate that the inpainting algorithm in this study can effectively eliminate the occlusion of normal and abnormal tissues caused by thermoelectric devices, enabling the normal visual observation function of the endoscope. In addition, in the human intestinal model (Fig. 4c(i)), we also conduct fast and slow contact tests on normal tissues: Fig. 4c(ii) and (iii) respectively show the comparison of normal force and tangential force sensed by the T-scope probe during fast contact and slow contact. Experimental results show that the increase rate of normal force (-1.4 vs -0.17) and the vibration amplitude of lateral force (-58 mN vs 21 mN) during fast contact are significantly greater than those during slow contact, which indirectly proves the sensitivity and accuracy of the 3D force sensing deep neural network in this study. For more detailed three-dimensional force sensing data, please refer to Supplementary Fig. S13 and S14, Movies S3, S4, and S5.

## In-vivo animal visual-tactile diagnosis verification experiment

To verify the efficacy of the T-scope system in real in-vivo scenarios, visual-tactile-temperature diagnostic experiments are conducted in the gastric cavity of anesthetized rabbits (Supplementary Movie S6). Prior to the experiments, a precise inflammation model of rabbit gastric tissue is established through inflammation modeling over a 4-day period. The T-scope probe is encapsulated at the front end of a handheld endoscope, which is then delivered into the rabbit's body through its oral cavity to perform visual-tactile endoscopic diagnosis (Fig. 5a). At the initial stage of the experiment, researchers navigate the probe delivery process using conventional endoscopic observation. Fig. 5b illustrates the visual observation sequence, which tracks the probe's path from the exterior of the rabbit's oral cavity, through the esophagus and cardia, and finally to the inflamed gastric tissue. This sequence not only demonstrates the system's excellent visual observation performance but also confirms the effectiveness and stability of the proposed AI visual restoration algorithm in real in-vivo scenarios (Supplementary Fig. S15).

During the pressing process on normal and abnormal tissues inside the living rabbit, the T-scope system is observed to achieve accurate visual reconstruction while exhibiting highly sensitive force estimation capabilities (Fig. 5c(i)). When researchers apply similar pressing forces to different types of tissues, it is found that the inflamed tissue—due to its higher hardness—exhibits significantly

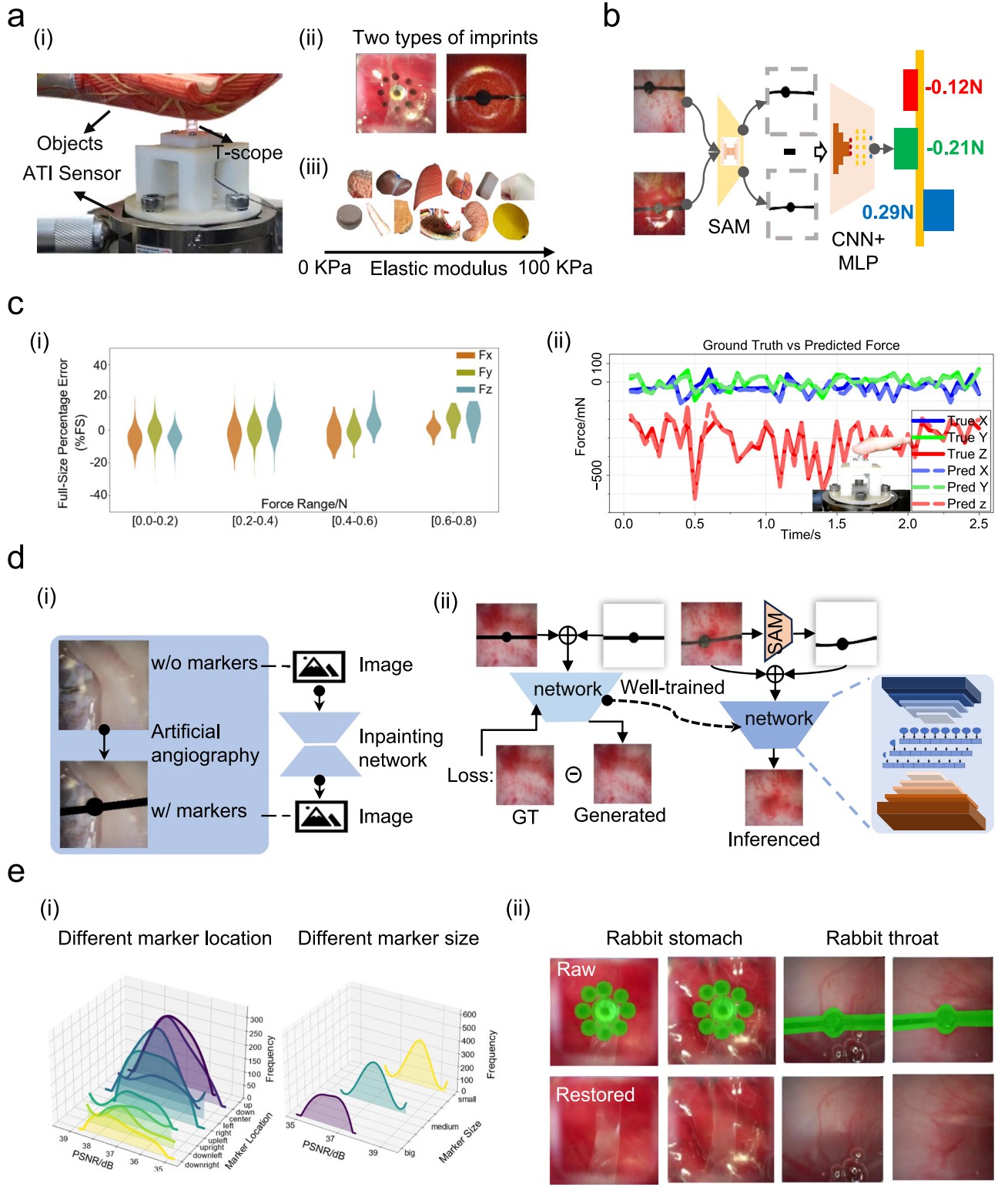

**Fig. 3 | AI-driven visual-tactile deep neural networks. ai** The tactile dataset collection setup. **aii** Two different types of thermoelectric imprints of the T-scope probe. **aiii** 18 soft materials used to collect 3 d force-tactile image samples, with hardness covering various types of human tissues from 0 to 100 KPa. **b** The EndoForce network (SAM: Segment Anything Model, CNN: Convolutional Neural Networks, MLP: MultiLayer Perceptron). **ci** The estimated force error distribution (full-scale error percentage, %FS) within different contact force ranges. **cii** Comparison of predicted force and ground-truth force when a person's finger randomly touches the probe. **di** The coating-free T-scope sensor to collect visual inpainting dataset. **dii** The training and inference process using the inpainting network. **ei** The restored Peak Signal-to-Noise Ratio (PSNR) distribution (decibels, dB) of different coating location and coating size. **eii** Visualization of visual inpainting results under various in-vivo scenarios. Source data are provided as a Source Data file.

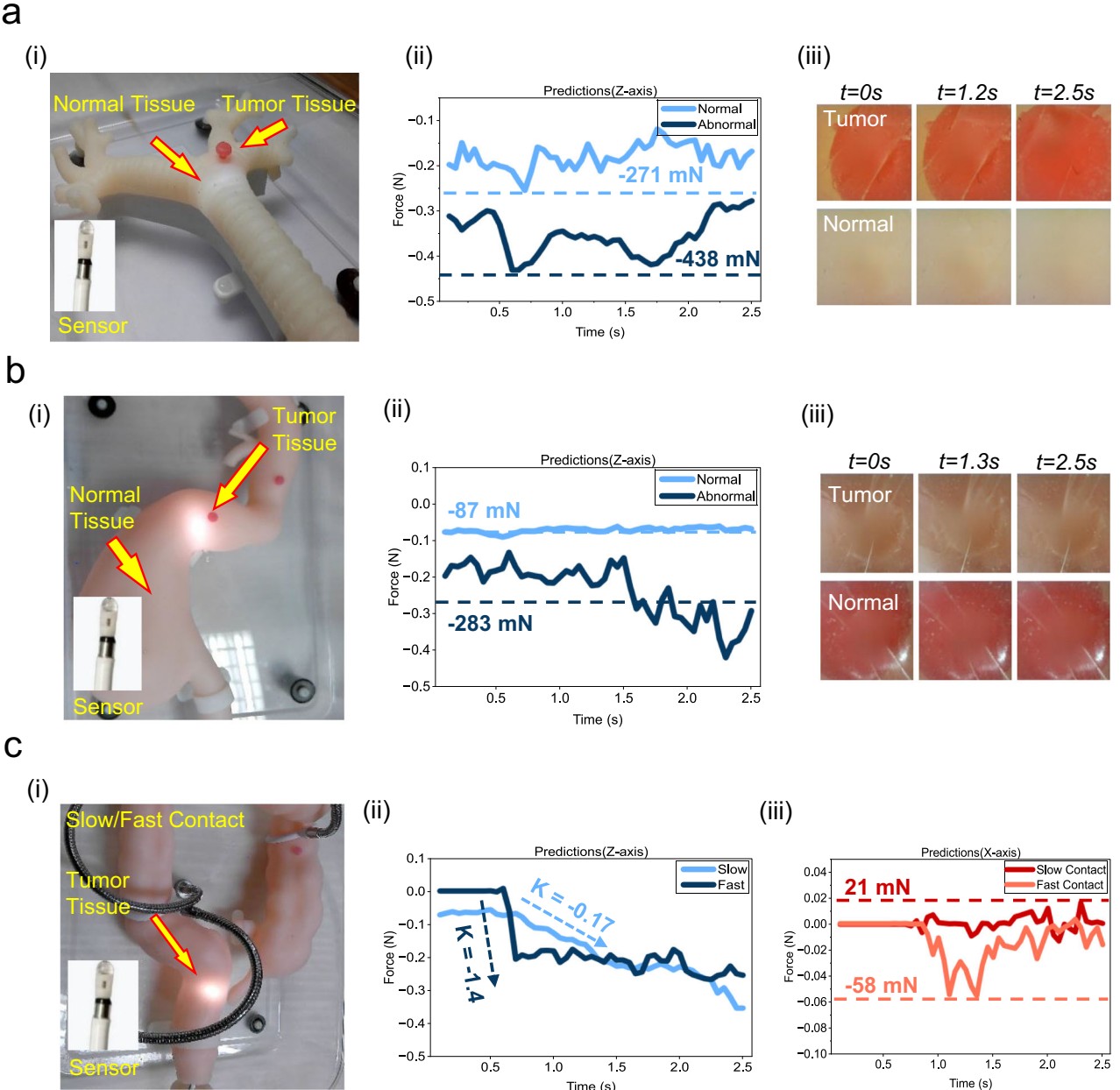

**Fig. 4 | In vitro contact experiment simulating human organ model. ai** Simulated bronchial tissue palpation is performed using a manually operated endoscope system equipped with a T-scope probe. **aii** Normal force sequences are measured in normal tissue and simulated tumor tissue at the same contact depth while the operator manually feeds at similar speeds. **aiii** Restoration of visual observation during contact (partial) in aii. **bi** Simulated stomach tissue palpation. **bii** Normal force sequences are measured in normal tissue and simulated tumor tissue at the same contact depth while stable contact. **biii** Restoration of visual observation during contact (partial) in bii. **ci** Simulated intestine tissue palpation. **cii** The z-axis force series in normal tissue and simulated tumor tissue are measured at the same contact depth while the operator manually feeds at fast and slow speeds. **ciii** The x-axis force series in normal tissue and simulated tumor tissue are measured at the same contact depth while the operator manually feeds at fast and slow speeds. Source data are provided as a Source Data file.

larger absolute values of normal force and a faster force increase rate compared to normal tissue (Fig. 5c(ii): rate, -0.016 vs -0.009; maximum absolute value of normal force, 516 mN vs 328 mN).

Furthermore, intensive contact sampling is performed in the vicinity of the inflamed tissue to obtain the temperature distribution of the inflamed tissue and its boundary regions (Fig. 5d(i)). The results reveal that the temperature of the inflamed tissue is significantly higher than that of normal tissue, with a maximum temperature difference of approximately 4 °C. A sliding temperature measurement experiment is also conducted at the boundary of the inflamed tissue, where the probe slides from the normal tissue to the inflamed tissue.

The corresponding visual observations and temperature change curves are recorded (Fig. 5d(ii)). The results indicate that at the boundary of the inflamed tissue, a noticeable temperature rise occurs even when no obvious visual changes are observed. This finding confirms the significance of integrating temperature sensing into endoscopic systems, particularly for advancing precision medicine applications.

## Discussion

This study introduces T-scope, an endoscope platform that integrates optical, mechanical, and thermal sensing through polycrystalline

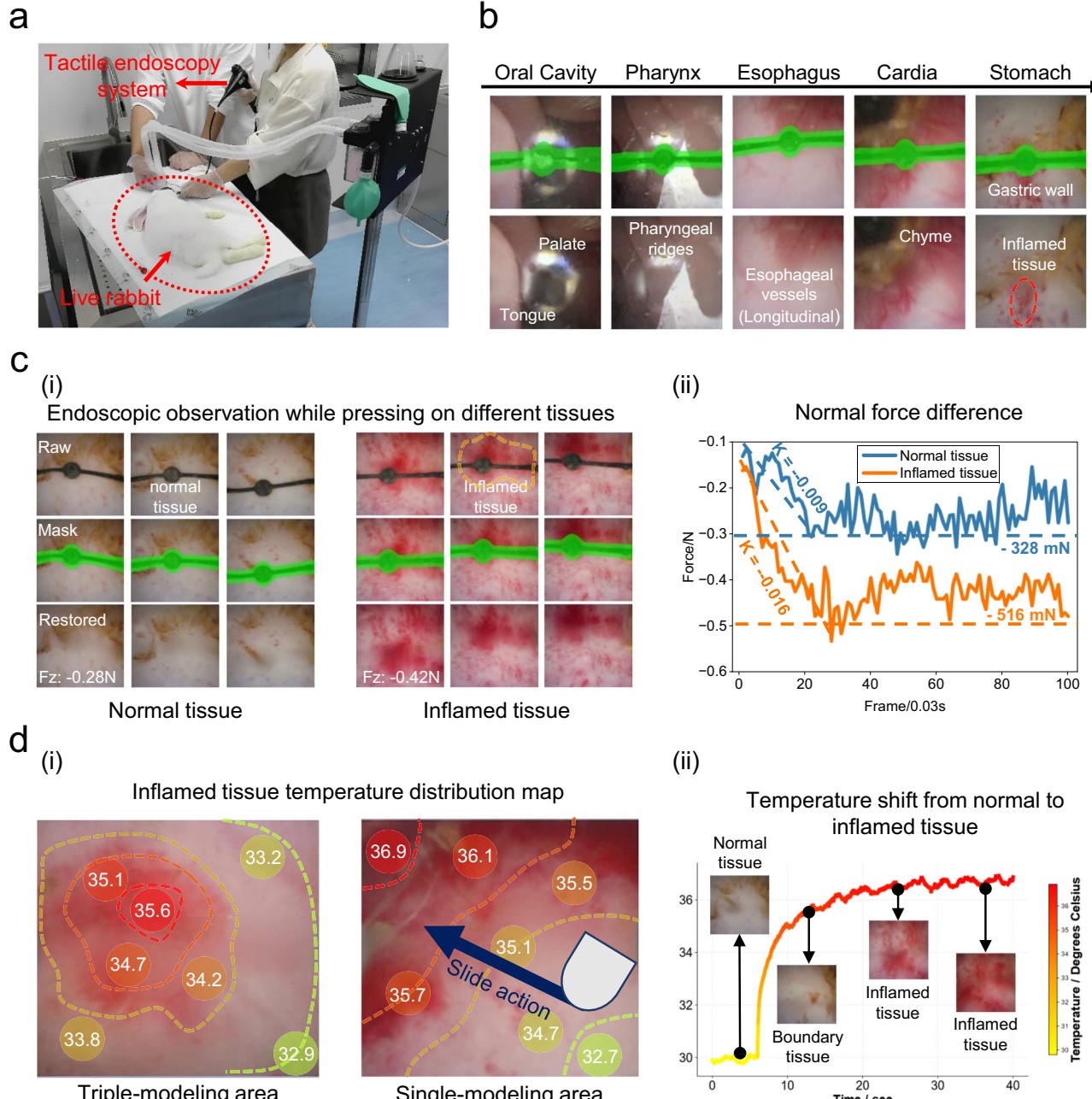

**Fig. 5 | Live rabbit visual-tactile diagnosis experiment. a** In-vivo contact experiment on rabbit gastric tissue using a manually operated T-scope probe. **b** Endoscopic visual navigation effect of the T-scope endoscopic system, starting from the oral cavity, passing through the esophagus and cardia, and finally reaching the inflamed tissue in the stomach. **ci** Comparison of normal force changes and visual observation results during the pressing process on normal gastric tissue and inflamed lesion tissue. **cii** Comparison chart of normal force change curves during two regular pressing processes on normal and inflamed tissues; the inflamed tissues are stiffer, resulting in greater speed change rate and pressure value. **di** Temperature distribution map obtained by multiple temperature measurements in the boundary area around the inflamed tissue. **dii** Temperature change curve and key frame images during the process of sliding the probe from the normal tissue to the inflamed tissue. Source data are provided as a Source Data file.

tellurium thin film technology, achieving integrated visual, force, and temperature tri-modal sensing at a spatial scale of 5 mm for the first time. Its core contribution lies in the engineered tellurium thin film matrix that enables simultaneous thermoelectric monitoring and strain-responsive force detection within a transparent microtextured elastomer probe. The platform combines machine vision and convolutional neural networks, using artificial intelligence-driven artifact suppression technology to directly relate tissue deformation patterns to force vectors while maintaining endoscopic imaging clarity, thus breaking through the limitations of traditional elastic mechanical modeling. Experimental validation confirms the consistency of T-scope's multimodal optical-mechanical-thermal measurements under controlled conditions.

We specifically discuss its performance and feasibility in three core dimensions: stability, biocompatibility, and cost. Stability: Our experimental results show that a single probe can reliably complete over 10k+ data acquisition cycles and support more than 20+ full animal experiments while maintaining operational functionality; to enhance structural stability, we design an integrated framework with both support and connection functions, and fully encapsulate the

entire structure with Ecoflex silicone. We also explicitly note that the probe currently lacks an additional overload-resistant structure and may be damaged under accidental force exceeding 5 N, but it is easily replaceable, and its multi-layer elastomer encapsulation ensures no components will remain in the body, effectively mitigating potential risks for practical use. Biocompatibility: The core materials of the T-scope system include Ecoflex elastomer and tellurium: Ecoflex has mature biocompatibility verification data to confirm its safety for biomedical scenarios; As for Te, it is tightly encapsulated by two inner rubber layers and an additional outer silicone coating. This multi-layer protection completely prevents Te from coming into direct contact with human tissue, further eliminating biocompatibility issues for translational use. Cost estimation: We provide a clear breakdown: the material cost for manufacturing a single probe is approximately 5 USD, a low-cost advantage that strongly supports the system's potential for large-scale practical application in clinical settings.

Future development will focus on developing an endoscopy system with tactile feedback, enabling operators to directly judge the hardness, temperature, and other conditions of the contacted tissue in real time through tactile force feedback from their hands, thereby improving diagnostic efficiency.

## Methods

### Mechanical structure overview of the T-scope probe

The miniaturized imaging module has a diameter of 3 mm (Supplementary Movie S1). This ultra-compact design enhances operational flexibility within confined anatomical spaces, enabling rapid navigation through complex tissue networks. To support this technology, the T-scope instrument maintains the critical dimensions of a 5 mm outer diameter and a 7.5 mm functional length, achieving an optimal balance between mechanical rigidity and space economy for endoscope compatibility. Supplementary Movies S3, S4, and S5, and S6 capture the system in action, demonstrating its ability to simultaneously acquire multimodal data during complex surgical procedures requiring millimeter-level precision.

### Thin-film processing of polycrystalline tellurium composites

As shown in Supplementary Fig. S3, the 10 μm polyethylene terephthalate substrate underwent sequential solvent purification: primary degreasing in ethanol via 30-minute ultrasonic bath followed by equivalent duration deionized water rinsing. To mitigate substrate deformation, the plasma-treated PET was electrostatically laminated onto polydimethylsiloxane backing. Selective metallization was achieved through removable adhesive masking, enabling magnetron-sputtered deposition of 50 nm Au films on 50% substrate area. Subsequent mask removal permitted full-surface tellurium encapsulation (50 nm thickness), establishing a bifunctional surface architecture with Au-Te heterojunction and pristine Te zones. Laser ablation ($\lambda = 1064$ nm) defined circular thermal transduction elements (1 mm diameter) at Au-Te/Te interfacial boundaries.

Precision electrode fabrication involved: (1) sputter-coating indium tin oxide through stepwise masked alignment, generating electrically isolated conductive traces; (2) laser-assisted micro-patterning to form peripheral interconnects. Final device integration coupled the thermal transducer with ITO electrode discontinuities through photolithographic alignment, completing assembly of the polycrystalline tellurium thermosensitive platform.

### Electrical Characterization of Te-based thermoelectric subsystem

We use a semiconductor analyzer (Keithley B1500A) to record the thermoelectric potential output of the sensing unit. We use a Peltier element as a single-side heat source, acquiring different temperature values through voltage control and calibrating with a thermocouple (UNI-T, UT325). To maintain the target contact temperature, we connect a thermocouple probe to the Peltier element and place a thermistor above the probe (Supplementary Note S4, Table S6). After applying a bias voltage to the Peltier element for 3 minutes, the corresponding temperature is recorded as the contact temperature. The opposite-side temperature is obtained by placing a Te-based thermistor between the thermocouple probe and the Peltier element. After applying a bias voltage to the Peltier element for 3 minutes, the corresponding temperature is recorded as the opposite-side temperature. To facilitate temperature calibration and enhance controllability, a commercially available temperature-controlled hot plate is subsequently used as the heat source to calibrate the temperature sensing subsystem within the integrated device.

### Elastomeric structural system with visuotactile encoding

The probe's load-bearing framework is constructed via additive manufacturing (layer resolution: 25 μm), achieving structural integrity within a 10 mm³ volumetric footprint. A transparent elastomeric dome (diameter: 5 mm, Shore 10 A compliance) is precision-molded using optical-grade silicone, engineered to simultaneously enable contact-induced strain visualization and maintain >92% light transmittance for endoscopic imaging. Tactile fiducial markers ($\varnothing = 200$ μm, carbon-particle enhanced) are strategically patterned to maximize tactile signal discrimination while maintaining <5% surface coverage for minimal optical obstruction.

Surface topology control is implemented through CNC-machined aerospace-grade aluminum molds ($R_a < 0.8$ μm), ensuring sub-micrometer surface roughness on the elastomer. The marker integration protocol involve: (1) micro-needle indentation ($\varnothing = 100$ μm) using vacuum-assisted mold transfer. This hierarchical fabrication methodology guarantees marker durability under cyclic contact stresses exceeding 50 kPa. Complete process schematics are provided in Supplementary Fig. S3.

### EndoForce dataset collection

To construct a reliable dataset for training the EndoForce network, researchers design a 3D force acquisition system (Supplementary Fig. S7). The endoscopic probe is fixed to an ATI Gamma force-torque transducer, with their coordinate systems precisely aligned. Initial calibration measurements show that the sensor exhibits baseline drift in the non-contact state. By controlling the interaction between the instrument tip and various soft tissues, researchers obtain synchronized recordings of triaxial force vectors and corresponding endoscopic images, which form the foundational training corpus for the machine learning framework.

The data acquisition protocol captures tissue deformation patterns in real time via the probe's imaging system, while simultaneously performing precise force measurements (the latter serving as reference values). The study employs 11 soft objects (with elastic modulus ranging from 0 to 100 kPa), covering the hardness range of real biological tissues, to verify mechanical performance in complex luminal environments.

Operators systematically apply contact forces at three levels (high, medium, and low) from 9 directions, ensuring coverage of a wide range of contact scenarios and force variations. To enhance the heterogeneity of the dataset, we fabricate 2 probes with different imprints under different operating environments. This methodology constructs a comprehensive 3D force database containing 121,800 experimental samples, of which 24,000 samples are used for network evaluation.

### EndoForce network

As depicted in Fig. 3b, our methodology initiates with acquiring spatial marker distribution data through an Efficient-SAM semantic segmentation model applied to tactile imagery. Subsequently, the system calculates the disparity between current segmented markers and their

baseline positions (recorded during non-interaction periods) to capture positional displacements.

Furthermore, an automated tracking system monitors centroid coordinates of the nine-marker array in real time. The system interprets stationary centroid coordinates as indicative of non-contact conditions, triggering a suspension of EndoForce network parameter updates until subsequent positional variations are detected. This design ensures computational efficiency during inactive phases while maintaining readiness for dynamic interaction analysis.

### Image inpainting dataset

To establish a comprehensive image inpainting dataset, we first design a modified T-scope sensor—this sensor removes the textured surface layer and enables the acquisition of clean tactile images without interference from physical markers. We use the T-scope to collect real tissue contact sequences in in-vivo animal experiments. Subsequently, we systematically superimpose artificial patterns that simulate real marker characteristics (including positional variation, dimensional diversity, and angular rotation) onto these baseline images. This method generates precisely aligned input-target image pairs, providing effective training conditions for the reconstruction algorithms used to remove patterns.

Through computational augmentation techniques detailed in Supplementary Fig. S10, we generate multiple synthetic variants of each original image that match the distribution of ground-truth labels, enabling image expansion. This process ultimately constructs a large-scale dataset containing 17,500 samples, of which 900 samples are reserved for the final evaluation of the network.

### Visual inpainting network

The raw input undergoes feature extraction through the segmentation framework, which precisely localizes the geometric distribution of marker patterns. To enhance reconstruction integrity, the algorithm expands the initial segmentation boundaries by 20% of the original dimensions, generating an enhanced mask that incorporates potential interference regions. This mask serves as a critical parameter to guide subsequent inpainting operations, effectively eliminating visual artifacts induced by markers. The trained inpainting network processes both the tactile image and augmented mask in parallel, ultimately producing a high-fidelity tissue surface representation devoid of marker interference.

### Computational latency profiling

The current prototype prioritizes functional validation over runtime optimization. The image acquisition subsystem employs a Raspberry Pi 4B microcomputer executing Python-based capture routines, transmitting 224×224 pixel data at 30 fps through Gigabit Ethernet to a host workstation equipped with a GeForce RTX 4080 GPU (16GB VRAM). Deep learning models for 3D force field prediction and image restoration achieve respective throughputs of 30 fps and 17.8 fps, meeting real-time interactive requirements.

### In vivo experimental protocol

All experimental procedures comply with the ethical guidelines are approved by the Wuhan Laboratory Animal Administration Office (Protocol ID: HZAURAB-2025-0017). Please refer to Supplementary Note S5 for the specific implementation process.

## Data availability

The raw data generated in this study have been deposited in the database under https://doi.org/10.5281/zenodo.18041502. All data are available from the corresponding authors upon request. Source data are provided in this paper.

## Code availability

The code is available on GitHub (https://github.com/tryallfailure/T-Scope). The DOI for this code is https://doi.org/10.5281/zenodo.18041502. This repository provides the complete implementation, datasets, and results for the T-Scope system.

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

## Acknowledgements

This work was supported by the National Key Research and Development Program of China under 2024YFB3212100 (L.W.); the Beijing Natural Science Foundation under Grant L253006 (S.C.); the National Natural Science Foundation of China under Grant 62025307 (L.C.), 62303455 (S.C.), U23B2038 (S.W.), U24A20282 (Y.W.), U25A20455 (T.Z.) and 62422409 (L.W.); and the China National Postdoctoral Program for Innovative Talents BX20240360 (L.L.).

## Author contributions

S.C., L.W. and Y.W. designed the research. S.C., L.L., Z.H. and Y.W. wrote the paper. S.C., L.L., Y.Y., X. B., C.Z., T.Z., L.C., W. Z., M.C. and Y.W. performed the experiments. S.C. performed the first principles calculations and simulations. S.C., X. B., Z.H. and L.L. analyzed the data. S.W., Z.L., W.G, S.W., C.W., L.W. and Y.W. revised the paper. C.W., L.W. and Y.W. supervised the project. All authors substantially contributed to the research and reviewed the manuscript.

## Competing interests

The authors declare no competing interests.
