## [Transparent Peer Review file · Nature Communications]

Superelastic Tellurium Thermoelectric Coatings for Advanced Trimodal Microsensing

Corresponding Author: Professor Yu Wang

Version 0:

Reviewer comments:

Reviewer #1

(Remarks to the Author)
NCOMMS-25-66562 - Peer-Review

The manuscript "Superelastic Tellurium Thermoelectric Coating for Advanced Trimodal Microsensing" is meaningful because it simultaneously implements visual, tactile, and temperature sensing in a single endoscopic probe. However, compared to several previously published studies on the visuo-tactile endoscope, there are still many shortcomings, and the temperature measurement does not significantly impact in its performance. It is an example of applying the additional benefits of the material's properties to the endoscope.

Even in the existing published literature - for example, the Vision-based Tactile Sensor for Endoscopy reported by Takashima et al. (DOI: 10.1007/978-4-431-30962-8_2) attempts have been made to provide contact information along with visual images by integrating tactile sensors at the end of the endoscope. This work has shown the potential to provide real-time tactile feedback with minimal degradation in the quality of optical images. One of the author's basic ideas for measuring tactile force in your paper seems to originate here, which is surprising that it is not a reference.

One of the biggest issues with the proposed study was the study of endoscopes, which were only verified at a level that seemed significantly less likely to be used, and even if they were clinically verified, the performance suggested is still limited. The effectiveness of the study is minimal because the tumor simulation used in the study is only a simplified material, and the experiments on rabbit and pig stomach tissues do not reflect the various pathological complexities that appear in the actual patient environment.

AI models are also lacking. EndoForce and ProPainter networks have been utilized for 3D force estimation and image correction (impainting), respectively, but the dataset is gathered in a very limited environment, resulting in insufficient generalization to real external environments.

Overall, it seems to be nothing more than a study that finds application cases of special sensor materials, and the research results are comprehensively insufficient to be published in Nature Communications, so rejection of publication seems appropriate.

Additional review contents have been added to the following.

1. The main text requires descriptions for Figures 2c, 2e, 3c, 4e, 4f, and 4g. These figures are present, but their corresponding explanations are missing from the body of the paper.
2. In lines 223-225, Figures 2i and 2j are explained together. However, it would be better to provide a separate and more detailed explanation for Figure 2i. Based on the current sentence, it is difficult to understand what Figure 2i represents.
3. In line 274, the statement, "The marker layer enabling tactile perception introduces visual occlusion in endoscopic images (Fig. 3f)," does not align well with the content of Figure 3f. A more appropriate description or a more suitable figure is needed.
4. Overall, the figures in the "Results" section are not well described. The figures themselves are difficult to understand and should be replaced with more effective ones. Additionally, the descriptions for all figures need to be thoroughly reviewed and revised.

Reviewer #2

(Remarks to the Author)

This manuscript presents a novel multimodal endoscopic sensing platform, the T-scope, which integrates visual, tactile, and thermal sensing into a single functioning device. The work addresses a longstanding technological limitation of conventional endoscopes, namely their inability to provide tactile and temperature perception alongside visual imaging. The authors report a technically innovative solution using tellurium thermocouples and viscoelastic silicone encapsulation, further enhanced by optimized heterointerfaces and deep neural network algorithms. As a result, the device is capable of allowing high-quality imaging, precise thermal mapping, and force feedback at microscale resolution. Moreover, the T-scope is compatible with standard and commercial endoscopes, and clinical verification has been performed through in vivo and in vitro experiments. The manuscript is well-structured and technically rigorous; however, a few minor revisions would further enhance its overall clarity and readability.

- 1) The introduction would benefit from a clearer comparison with existing benchmarks and state-of-the-art approaches. In particular, an explicit discussion of tellurium relative to alternative materials would strengthen the rationale for selecting this material.
- 2) Further discussion on the stability, biocompatibility, and cost estimation of the proposed system would be valuable to assess its translational potential.
- 3) A more detailed description of the experimental setup for temperature measurements would enhance transparency and reproducibility. Have the authors employed a thermocouple as a reference instrument to acquire the temperature?
- 4) The readability of Figures 2 and 3 is limited; incorporating a legend and clearer annotations would substantially improve their interpretability.
- 5) The authors are encouraged to discuss whether, in addition to visual feedback, the integration of alternative modalities—such as advanced haptic feedback systems—could further strengthen the closed-loop performance of the instrument. Including such considerations in the discussion section would highlight opportunities for future development.

Reviewer #3

(Remarks to the Author)

This work proposes a tellurium-based thermoelectric visual-tactile endoscope that integrates imaging, temperature sensing, and force feedback into a single platform. While the system is promising for multifunctional endoscopic applications, some questions remain to be addressed.

1. The definitions of 'contact beginning,' 'stable contact,' 'fast contact,' and 'slow contact' are ambiguous. This makes it difficult to interpret the results of the in vitro contact experiment. Please clearly define these terms, for example, by specifying them along the time axis, to improve the clarity of the analysis.
2. Some figures included in the main manuscript are not cited in the text. Please check all figure references to ensure every figure is properly called out and discussed.
3. A significant number of figures contain a variety of typographical and display errors. For instance, in Figure 4i, the label "contact beginning" is improperly positioned and overlaps with other text. It is highly recommended that the authors perform a comprehensive and meticulous review of all figures, including the supplementary figures, to correct these issues. Ensuring the accuracy of all values, labels, and legends is crucial for the credibility of the presented data.

Version 1:

Reviewer comments:

Reviewer #1

(Remarks to the Author)

1. In Figure 3 c(i), the error range for z error appears significantly larger compared to x error and y error. Please explain why the error range is large only for the z-axis.
2. The accuracy of the z-axis force seems most critical for measuring contact force. However, in Figure 3 c(i, ii) of this paper, the z error shows the largest deviation. Can a system with comparatively less accurate z error (compared to x error and y error) be used for a tactile endoscope? Please explain whether this error range is acceptable by comparing it with previous research.
3. On lines 323-324, the paper states, "we conduct ex vivo experiments using full-scale biosimulation phantoms." It seems that actual biological tissues were not used, yet the term "ex vivo" was. Is the use of the term "ex vivo" appropriate here?

Please explain this.

Reviewer #2

(Remarks to the Author)

I have carefully reviewed the revised manuscript along with the authors' point-by-point responses. All major issues and suggestions raised in my initial review have been thoroughly and convincingly addressed. The revisions have notably strengthened the manuscript, particularly through the improved clarity and depth of the Introduction and Discussion sections. The scientific rigor, coherence, and overall presentation have been substantially enhanced. Furthermore, the figures have been made more readable, with clearer descriptions and labeling that eliminate previous ambiguities. I am fully satisfied with the current version and have no additional comments.

I therefore recommend the manuscript for acceptance in its present form.

Reviewer #3

(Remarks to the Author)

The authors have addressed all the comments. I recommend the publication on Nature communications.

Version 2:

Reviewer comments:

Reviewer #1

(Remarks to the Author)

I have no further comments and recommend this manuscript for acceptance.

Author Response to Reviews of

Superelastic Tellurium Thermoelectric Coatings for Advanced Trimodal Microsensing (NCOMMS-25-66562)

Shaowei Cui, Linlin Li, Zi-xin Huang, Yanzhe Yu, Mingxue Cai, Xiangyin Bao, Chaofan Zhang, Tiandong Zhang, Long Cheng, Wenxuan Zhang, Zheng Lou, Shuo Wang, Wen Gong, Chao-feng Wu, Lili Wang, Yu Wang

RC: Reviewer Comment, AR: Author Response

Reviewer #1

General Comment

RC: *The manuscript "Superelastic Tellurium Thermoelectric Coating for Advanced Trimodal Microsensing" is meaningful because it simultaneously implements visual, tactile, and temperature sensing in a single endoscopic probe. However, compared to several previously published studies on the visuo-tactile endoscope, there are still many shortcomings, and the temperature measurement does not significantly impact in its performance. It is an example of applying the additional benefits of the material's properties to the endoscope.*

AR: Many thanks for your professional comments. Compared with previous tactile endoscope studies, our study focus on **introducing temperature sensing into a visual-tactile endoscopy system, achieving simultaneous multimodal sensing (visual, force and temperature) on a common endoscopy probe for the first time.**

Firstly, we confirmed the significance of the temperature modality through consultations with practicing physicians. Additionally, during our review of existing research, **temperature is a crucial monitoring indicator in endoscopic surgery [1,2]**. Kozen *et al.* mentioned that "Although the theoretical risk of elevated temperatures during endoscopic ear surgery has been reported previously, neither temperature change nor heat distribution associated with the endoscope has been quantified [2]." Meanwhile, **temperature is also important for precision medical diagnosis of lesional tissues [3,4]**. Kim *et al.* mentioned that "In optical imaging of solid tumors, signal contrasts derived from inherent tissue temperature differences have been employed to distinguish tumor masses from surrounding tissue [3]."

References:

[1] Madhvapathy S R, Bury M I, Wang L W, et al. Miniaturized implantable temperature sensors for the long-term monitoring of chronic intestinal inflammation[J]. *Nature Biomedical Engineering*, 2024, 8(8): 1040-1052.

[2] Kozin E D, Lehmann A, Carter M, et al. Thermal effects of endoscopy in a human temporal bone model: implications for endoscopic ear surgery[J]. *The Laryngoscope*, 2014, 124(8): E332-E339.

[3] Kim S, Oh G, Kim Y R, et al. Infrared thermal modulation endoscopy for label-free tumor detection[J].

Scientific Reports, 2024, 14(1): 31575.

[4] Calvarese M, Corbetta E, Contreras J, et al. Endoscopic AI-driven morphochemical imaging and fs-laser ablation for selective tumor identification and selective tissue removal[J]. Science Advances, 2024, 10(50): eado9721.

However, while the aforementioned studies have demonstrated the significance of temperature in tissue diagnosis, **most existing studies have used independent channel temperature sensing systems and have not yet integrated these technologies into endoscopy systems.** To the best of our knowledge, the T-scope system proposed by us is the first trimodal endoscopic sensing system that enables direct temperature measurement at the contact tip while providing real-time feedback on both force and original visual observation.

In our revision, the results of endoscopic diagnosis experiments on living animals indicate that at the boundary of inflamed tissue, even though the visual modality did not observe abnormal changes compared with normal tissue, the temperature showed a significant increase. This finding confirms the significance of integrating temperature sensing into endoscopic systems, particularly for advancing precision medicine applications. Thank you again for your professional comments, which have greatly improved the quality of our manuscript.

In the revised manuscript, we have comprehensively revised the abstract, introduction, AI model, and animal studies. We have selected some key revisions as follows:

Revision:

INTRODUCTION: " **However, a notable limitation of current tactile endoscope systems lies in their general lack of temperature-sensing functionality, which is a critical capability for endoscopic surgeries.** The involvement of temperature modalities can achieve necessary thermal monitoring^{10,11} (real-time target tissue temperature measurement to avoid thermal damage) and play a key role in the accurate diagnosis of abnormal tissues¹²⁻¹⁴ (such as identifying inflammation and tumor lesions through local temperature changes).

RESULTS (2.6): "To verify the efficacy of the T-scope system in real in-vivo scenarios, visual-tactile-temperature diagnostic experiments are conducted in the gastric cavity of anesthetized rabbits (Supplementary Movie S6). Prior to the experiments, **a precise inflammation model of rabbit gastric tissue is established through inflammation modeling over a 4-day period.**"

"A sliding temperature measurement experiment is also conducted at the boundary of the inflamed tissue, where the probe slides from the normal tissue to the inflamed tissue. The corresponding visual observations and temperature change curves are recorded (Fig. 5d (ii)). **The results indicate that at the boundary of the inflamed tissue, a noticeable temperature rise occurs even when no obvious visual changes are observed. This finding confirms the significance of integrating temperature sensing into endoscopic systems, particularly for advancing precision medicine applications.**"

Major Comment #1

RC: *Even in the existing published literature - for example, the Vision-based Tactile Sensor for Endoscopy reported by Takashima et al.(DOI: 10.1007/978-4-431-30962-8_2) attempts have been made to provide contact information along with visual images by integrating tactile sensors at the end of the endoscope. This work has shown the potential to provide real-time tactile feedback with minimal degradation in the quality of optical images. One of the author's basic ideas for measuring tactile force in your paper seems to originate here, which is surprising that it is not a reference.*

AR: Thank you very much for your valuable comment, which has helped us identify an important oversight in the initial version of our manuscript. We sincerely apologize for failing to properly cite the significant work by Takashima et al. (DOI: 10.1007/978-4-431-30962-8_2) on the "Vision-based Tactile Sensor for Endoscopy" in the first draft.

In the revised manuscript, we have made targeted supplements and improvements:

- 1) We have formally cited Takashima et al.'s work in the corresponding section, and have added a detailed discussion to clarify the connection and distinction between our research and this study;
- 2) We have further sorted out and summarized the latest research progress in the field of tactile endoscopy, and added a comparative analysis with multiple representative studies (including this work) to more systematically present the positioning and innovation of our research in the existing academic context.

Once again, we appreciate your careful review and critical feedback, which has significantly helped improve the completeness and rigor of our manuscript.

Revision:

ABSTRACT: "Tactile endoscopes can provide physicians with rich sensory information, enabling fast and accurate medical diagnoses. **However, existing tactile endoscopy sensors do not consider temperature perception, which is a very important diagnostic indicator in medicine.**"

INTRODUCTION: "In the past decades, tactile endoscope systems have achieved force measurement while maintaining basic visual observation capabilities³⁻⁶. This marks progress in enhancing the safety and accuracy of endoscopic procedures by enabling surgeons to obtain both visual insights into the surgical site and quantitative force feedback⁷⁻⁹. However, a notable limitation of current tactile endoscope systems lies in their general lack of temperature-sensing functionality, which is a critical capability for endoscopic surgeries. The involvement of temperature modalities can achieve necessary thermal monitoring^{10,11} (real-time target tissue temperature measurement to avoid thermal damage) and play a key role in the accurate diagnosis of abnormal tissues¹²⁻¹⁴ (such as identifying inflammation and tumor lesions through local temperature changes). Therefore, how to seamlessly integrate temperature sensing technology into tactile endoscopy systems to achieve high-precision 3D force measurement and high-definition visual imaging without compromising the system's existing compact structure remains a significant challenge in this field. "

REFERENCE: "3. Takashima, K., Yoshinaka, K., Ikeuchi, K.: Vision-based tactile sensor for endoscopy. In: Complex Medical Engineering. Springer Japan, Tokyo, pp. 13–23 (2007)"

Major Comment #2

RC: *One of the biggest issues with the proposed study was the study of endoscopes, which were only verified at a level that seemed significantly less likely to be used, and even if they were clinically verified, the performance suggested is still limited. The effectiveness of the study is minimal because the tumor simulation used in the study is only a simplified material, and the experiments on rabbit and pig stomach tissues do not reflect the various pathological complexities that appear in the actual patient environment.*

AR: Thank you very much for your valuable comments and constructive suggestions, which have provided important guidance for improving our study. Our ultimate goal is undoubtedly to apply the T-scope system to human endoscopy diagnosis, but due to the review restrictions of human experiments and the research

stage, we are unable to conduct real human experiments at this stage. **Fortunately, we can refer to existing experimental methods of existing endoscopic diagnostic literature, which use mammals such as pigs, rabbits, dogs, and mouse [1, 2, 3, 4, 5, 6] to simulate complex human lesions.**

References:

- [1] Kong W, Sun B, Wang Z, et al. Physiologically based pharmacokinetic–pharmacodynamic modeling for prediction of vonoprazan pharmacokinetics and its inhibition on gastric acid secretion following intravenous/oral administration to rats, dogs and humans[J]. *Acta Pharmacologica Sinica*, 2020, 41(6): 852-865.
- [2] Long Y, Lin A, Kwok D H C, et al. Surgical embodied intelligence for generalized task autonomy in laparoscopic robot-assisted surgery[J]. *Science Robotics*, 2025, 10(104): eadt3093.
- [3] Shademan A, Decker R S, Opfermann J D, et al. Supervised autonomous robotic soft tissue surgery[J]. *Science translational medicine*, 2016, 8(337): 337ra64-337ra64.
- [4] Saeidi H, Opfermann J D, Kam M, et al. Autonomous robotic laparoscopic surgery for intestinal anastomosis[J]. *Science robotics*, 2022, 7(62): eabj2908.
- [5] Garrick T, Mulvihill S, Buack S, et al. Intracerebroventricular pressure inhibits gastric antral and duodenal contractility but not acid secretion in conscious rabbits[J]. *Gastroenterology*, 1988, 95(1): 26-31.
- [6] Go G, Jeong S G, Yoo A, et al. Human adipose–derived mesenchymal stem cell–based medical microrobot system for knee cartilage regeneration in-vivo[J]. *Science Robotics*, 2020, 5(38): eaay6626.

We found that pigs and rabbits are the most commonly used laboratory animals. For instance, pigs were employed as experimental subjects for autonomous robotic surgery in [3], while magnetically controlled surgery was performed on rabbits in [6]. We finally chose live domestic rabbits—a more space-challenging option—as the laboratory animals for conducting in-vivo experiments in the revised manuscript.

Taking full account of the pathological complexity of human lesion tissues, we ultimately chose to induce inflammation to establish a model of the gastric tissue in experimental rabbits over a 4-day period to simulate the actual patient environment.

Specifically, we established an inflammation model in the stomach of live rabbits and conducted a complete visual-tactile endoscopic palpation experiment using the proposed T-scope haptic endoscope system. These experiments covered three complete processes: visual navigation, visual observation, temperature and force sensing of inflamed tissues. In addition to the original endoscopic visual observation function, we achieved accurate tactile diagnosis and obtained information on the temperature distribution and mechanical response characteristics of the entire inflamed tissue. **The experimental results show that even when no obvious visual changes are observed, a significant temperature increase occurs at the boundary of inflamed tissues. This finding confirms the importance of integrating temperature sensing technology into endoscopic systems, which is particularly crucial for advancing precision medicine applications.**

Revision:

RESULTS (2.6): To verify the efficacy of the T-scope system in real in-vivo scenarios, visual-tactile-temperature diagnostic experiments are conducted in the gastric cavity of anesthetized rabbits (Supplementary Movie S6). Prior to the experiments, a precise inflammation model of rabbit gastric tissue is established through inflammation modeling over a 4-day period. The T-scope probe is encapsulated at the front end of a handheld endoscope, which is then delivered into the rabbit's body through its oral cavity to perform visual-tactile endoscopic diagnosis (Fig. 5a). At the initial stage of the experiment, researchers navigate the

Fig. 5 | Live rabbit visual-tactile diagnosis experiment. **a** in-vivo contact experiment on rabbit gastric tissue using a manually operated T-scope probe. **b** Endoscopic visual navigation effect of the T-scope endoscopic system, starting from the oral cavity, passing through the esophagus and cardia, and finally reaching the inflamed tissue in the stomach. **c** (i) Comparison of normal force changes and visual observation results during the pressing process on normal gastric tissue and inflamed lesion tissue. (ii) Comparison chart of normal force change curves during two regular pressing processes on normal and inflamed tissues; the inflamed tissue is stiffer, resulting in greater speed change rate and pressure value. **d** (i) Temperature distribution map obtained by multiple temperature measurements in the boundary area around the inflamed tissue. (ii) Temperature change curve and key frame images during the process of sliding the probe from the normal tissue to the inflamed tissue.

probe delivery process using conventional endoscopic observation. Fig. 5b illustrates the visual observation sequence, which tracks the probe's path from the exterior of the rabbit's oral cavity, through the esophagus and cardia, and finally to the inflamed gastric tissue. This sequence not only demonstrates the system's excellent visual observation performance but also confirms the effectiveness and stability of the proposed AI visual restoration algorithm in real in-vivo scenarios (Supplementary Fig. S15).

During the pressing process on normal and abnormal tissues inside the living rabbit, the T-scope system

is observed to achieve accurate visual reconstruction while exhibiting highly sensitive force estimation capabilities (Fig. 5c (i)). When researchers apply similar pressing forces to different types of tissues, it is found that the inflamed tissue—due to its higher hardness—exhibits significantly larger absolute values of normal force and a faster force increase rate compared to normal tissue (Fig. 5c (ii): rate, -0.016 vs -0.009; maximum absolute value of normal force, 516 mN vs 328 mN).

Furthermore, intensive contact sampling is performed in the vicinity of the inflamed tissue to obtain the temperature distribution of the inflamed tissue and its boundary regions (Fig. 5d (i)). The results reveal that the temperature of the inflamed tissue is significantly higher than that of normal tissue, with a maximum temperature difference of approximately 4 °C. A sliding temperature measurement experiment is also conducted at the boundary of the inflamed tissue, where the probe slides from the normal tissue to the inflamed tissue. The corresponding visual observations and temperature change curves are recorded (Fig. 5d (ii)). The results indicate that at the boundary of the inflamed tissue, a noticeable temperature rise occurs even when no obvious visual changes are observed. This finding confirms the significance of integrating temperature sensing into endoscopic systems, particularly for advancing precision medicine applications.

Major Comment #3

RC: *AI models are also lacking. EndoForce and ProPainter networks have been utilized for 3D force estimation and image correction (impainting), respectively, but the dataset is gathered in a very limited environment, resulting in insufficient generalization to real external environments. Overall, it seems to be nothing more than a study that finds application cases of special sensor materials, and the research results are comprehensively insufficient to be published in Nature Communications, so rejection of publication seems appropriate.*

AR: Thank you very much for your professional comments. **Regarding the ProPainter network, in our revised manuscript, we have comprehensively replaced the previous simulated tissue acquisition protocol with marker-free visual image acquisition conducted on the laryngeal and gastrotracheal regions of live animals.** The entire dataset comprises approximately 65k data samples, and we performed a generalization test in real live animal experiments, which involved visual observation of various types of normal and pathological tissues (including inflammation and wounds). The results can be found in Supplementary Fig. S15 of our revised manuscript.

In terms of the tactile AI model, the proposed EndoForce model extracts Te imprint masks directly from raw images for deformation tracking. **The force estimation network is independent of the visual background of real visceral tissues, but highly correlated with the hardness of the contacted material.** Therefore, we have significantly expanded the data collection scenarios for the tactile network, no longer limiting it to simulated or real organs. **In the original manuscript, we used a simulated human stomach model for force data collection. In the revised manuscript, we instead collected data on real animal tissues, including 11 materials covering a wide range of stiffness (including real pig stomach, pig liver, pig viscera, pig tongue, human skin and other real animal organ tissues), covering the hardness range of common human soft tissues (spanning 0–100 kPa [1]).** For reference, the hardness of human gastric tissue typically ranges from approximately 0.5 to 5 kPa [2].

References:

[1] Hai P, Zhou Y, Gong L, et al. Quantitative photoacoustic elastography of Young's modulus in humans[C]//Photons Plus Ultrasound: Imaging and Sensing 2017. SPIE, 2017, 10064: 34-41.

[2] Jang M, An J, Oh S W, et al. Matrix stiffness epigenetically regulates the oncogenic activation of the

Yes-associated protein in gastric cancer[J]. Nature Biomedical Engineering, 2021, 5(1): 114-123.

Meanwhile, in our in-vivo experiments, we used the T-scope to make repeated contact with various types of real gastric tissues and abnormal tissues, with the number of contact instances exceeding 50. These tissues typically exhibit distinct hardness distributions, which result in a wide range of force-sensing characteristics—as illustrated in Fig. 5 of the revised manuscript. The results demonstrate the effectiveness of our EndoForce in force estimation for different real tissues.

Revision:

RESULTS (2.4): "Because the proposed thermoelectric devices provide natural landmarks in the endoscopic field of view, our endoscopic vision system can capture their position and overall shape changes, thereby creating an accurate map of the 3D contact forces acting on the probe^{36,37}. We first design a tactile data acquisition system (Fig. 3a(i)): two T-scope tactile endoscopic probes with different marker patterns (Fig. 3a(ii)) are fixed on a commercial multi-dimensional force sensor. Contact tests with varying directions and magnitudes are conducted using a variety of soft materials that fully cover the hardness range of human tissues (elastic modulus from 0 to 100 kPa) (Fig. 3a(iii), Supplementary Fig. S7). The six-axis force/torque sensor provides the ground-truth 3D forces acting on the surface of the tactile probe. Subsequently, we adopt a data-driven approach and propose the EndoForce 3D force estimation network to directly estimate 3D forces from raw visual observation inputs."

Supplementary Fig. S7 | The 3D force dataset collection pipeline. **a** The T-scope sensor is fixed on an standard ATI six-axis force/torque sensor, and a real piece of pig stomach tissue is contacted with the sensor surface manually. **b** A dozen objects covering the hardness range of 0-100KPa are used for data collection. **c** The 3D force dataset contains abundant tactile image-force pairs, and each sample provides a ground-truth 3D force label under various contact scenes.

RESULTS (2.4): "Thermoelectrical imprints cause visual occlusion in endoscopic images (Fig. 3a(ii)), which may compromise the clarity of the diagnostic field of view. To address this issue, we adopt ProPainter³⁹ for AI-driven visual inpainting. We first prepare a dataset suitable for real in-vivo tissue scenarios (Fig. 3d(i), Supplementary Note S3). Specifically, we use a T-scope probe without thermal-electrical markers to capture videos of the respiratory tract and gastric tissues of live rabbits. Visual inpainting synthetic datasets are then generated by manually adding various markers, including different dot arrays and dot-line patterns (Supplementary Fig. S10)."

Supplementary Fig. S10 | An example of the artificial angiography process.

RESULTS (2.4): "We first retrain the ProPainter network using the synthetic dataset with ground-truth labels. In real in-vivo experiments, the visual inpainting network with the trained weights is directly used for real-time visual observation (Fig. 3d(ii)). Results from the in-vivo experiments show that the cross-domain trained model successfully reconstructs visual information occluded by different thermoelectrical imprints (Fig. 3e(ii)). More inpainting examples can be found in Supplementary Fig. S11 and S12."

Supplementary Fig. S12 | Visualization of an image restoration example. We place a probe with a dot-line array imprint on the oral cavity, esophagus, and stomach of a living rabbit, generating visual images and the restoration results.

Minor Comment #1

RC: *The main text requires descriptions for Figures 2c, 2e, 3c, 4e, 4f, and 4g. These figures are present, but their corresponding explanations are missing from the body of the paper.*

AR: Thank you sincerely for your meticulous review and valuable feedback, which has helped us identify an important oversight in the manuscript. In the revised manuscript, we have supplemented detailed descriptive content for each of these six subfigures at their respective corresponding positions in the main text.

Please note that Figures 4e, 4f, and 4g in our original manuscript have been modified to Figures 4b(ii), 4b(iii), and 4c(i) in the revised manuscript.

Revision:

RESULTS (2.2): "This unique crystal structure exhibits significant lattice scattering, effectively suppressing lattice heat conduction and promoting the establishment of a significant temperature gradient (Fig. 2c). This enables efficient thermoelectric conversion and reduces the trade-off between thickness and responsivity. The fabrication process flow comprises the following key steps (Fig. 2d, Further details in the Methods section and Supplementary Fig. S3): Initial surface cleaning and PDMS composite treatment of 10 μm PET substrates; Magnetron sputtering deposition of gold electrodes on half-area substrates followed by full-substrate tellurium thin film deposition; Laser direct writing processing of AuTe-Te interfaces into 1 mm-diameter temperature-sensitive unit (Fig. 2e). When placed on a 42 $^{\circ}\text{C}$ hot plate, the temperature-sensitive unit demonstrates clear temperature field distribution characteristics on its surfaces, confirming temperature detection capabilities."

RESULTS (2.4): "Experimental results demonstrate that the system achieves excellent performance under varying contact magnitudes and directions (Supplementary Table S3 and S4). The average errors of the contact tangential and normal force components are 0.02 N and 0.06 N, respectively. Error distribution analysis reveals consistent prediction accuracy across different force magnitudes, with no error escalation observed when contact forces increase (Fig 3c(i)). Directional error analysis (Supplementary Fig. S9) indicates comparable performance across different orientations, though slightly higher errors occur in the central and downward directions. This phenomenon may be attributed to the contact habits of operators. Further validation through randomized finger contact trials (Fig. 3c(ii)) shows stable 3D force estimation with consistently low errors, confirming the generalization capability of the network."

RESULTS (2.5): "Biomechanical test data from the bronchial and gastric models (Fig. 4a(ii), b(ii)) shows that due to the higher biomechanical stiffness of tumor tissue, the fluctuation of its normal force value is more significant than that of healthy tissue when maintaining a stable contact state. Visual inpainting results (Fig. 4a(iii), b(iii)) indicate that the inpainting algorithm in this study can effectively eliminate the occlusion of normal and abnormal tissues caused by thermoelectric devices, enabling the normal visual observation function of the endoscope. In addition, in the human intestinal model (Fig. 4c(i)), we also conduct fast and slow contact tests on normal tissues: Fig. 4c(ii) and (iii) respectively show the comparison of normal force and tangential force sensed by the T-scope probe during fast contact and slow contact. Experimental results show that the increase rate of normal force (-1.4 vs -0.17) and the vibration amplitude of lateral force (-58 mN vs 21 mN) during fast contact are significantly greater than those during slow contact, which indirectly proves the sensitivity and accuracy of the 3D force sensing deep neural network in this study."

Minor Comment #2

RC: *In lines 223-225, Figures 2i and 2j are explained together. However, it would be better to provide a separate and more detailed explanation for Figure 2i. Based on the current sentence, it is difficult to understand what Figure 2i represents.*

AR: Thank you for your constructive suggestions. We deeply apologize for the confusing description in the original manuscript. Fig. 2i presents a simulation result at a contact temperature of 30 $^{\circ}\text{C}$. The image depicts the temperature measured on the opposite side of the sensor when it is exposed to a 30 $^{\circ}\text{C}$ heat source. This figure aims to illustrate the thermal conductivity differences between Te and AuTe more intuitively through numerical curves. We have revised the legend and description for Figure 2i in the revision to better convey the content and logic of the manuscript.

Revision:

Fig. 2 | Mechanism and evaluation of the thermoelectric subsystem. **a** The structure of the Te-based thermoelectric subsystem. **b** The lattice structure diagram of Te. **c** Thermal gradient under 42 °C thermal loading (Upper: Macroscopic observation; Lower: Infrared thermographic mapping). **d** Fabrication process flow for polycrystalline Te temperature detector (Magnetron sputtering thin-film deposition combined with laser-assisted micro-patterning). **e** A shot of the finished Te-based thermoelectric sensor. **f** Optical image of magnetron-sputtered ITO films deposited on flexible PET substrates, demonstrating high optical clarity and visible-light transmittance. **g** Computational simulation illustrating the working principle of thermoelectric sensing elements, incorporating an elastomeric contact layer at the top (to simulate the thermoelectric subsystem immobilized on the tactile interface) and a thermally regulated boundary at the bottom (to simulate different contact temperatures), with colormap visualization of the thermal distribution. The figure utilizes a contact temperature of 30 °C and an ambient temperature of 25 °C. **h** The seebeck-induced voltage mapping within the functional layers, quantitatively correlated with interfacial temperature modulation at the active contact region, 20-40 °C. **i** Distribution of thermal and electrical potentials under 30 °C contact temperature along the upper end of the subsystem. **j** Time-dependent voltage response characteristics of the device under varying thermal differentials, defined as the difference between the contact temperature and the ambient temperature. **k** Quantitative dependence of generated electrical potential on applied temperature.

RESULTS (2.3): "By applying thermal gradients (20-40 °C) at contact interfaces, simulations reveal significant temperature field redistribution and corresponding thermal potential variations (Fig. 2h, Supplementary Fig. S4). At a contact temperature of 30 °C, a 4 °C temperature gradient across the Te domain produces a 4 mV

output, demonstrating its robust thermal detection capabilities (Fig. 2i). The difference in thermal behavior between the Te and AuTe alloys is evident: AuTe maintains a uniform temperature distribution due to its high thermal and electrical conductivity, while Te's lower lattice thermal conductivity results in a 10°C temperature gradient at a 40 °C contact temperature. This contrast amplifies carrier accumulation on the Te surface, generating a 9 mV potential difference (Fig. 2h), confirming the mechanism of temperature-dependent signal generation.

Subsequently, the thermoelectric performance is evaluated under controlled contact heat sources (20–42 °C). The Te-based thermoelectric sensor exhibits stable output potential at identical contact temperatures, demonstrating its stability (Fig. 2j)."

Minor Comment #3

RC: *In line 274, the statement, "The marker layer enabling tactile perception introduces visual occlusion in endoscopic images (Fig. 3f)," does not align well with the content of Figure 3f. A more appropriate description or a more suitable figure is needed.*

AR: Thank you for your meticulous review and valuable feedback. We have comprehensively revised both Figure 3f and the corresponding text description. **Please note that Figure 3f in our original manuscript have been modified to Figure 3a(ii) in the revised manuscript.**

Revision:

RESULTS (2.4): "Thermoelectrical imprints cause visual occlusion in endoscopic images (Fig. 3a(ii)), which may compromise the clarity of the diagnostic field of view."

Minor Comment #4

RC: *Overall, the figures in the "Results" section are not well described. The figures themselves are difficult to understand and should be replaced with more effective ones. Additionally, the descriptions for all figures need to be thoroughly reviewed and revised.*

AR: Thank you sincerely for your critical and constructive comment, which directly points out a key weakness in the "Results" section of our manuscript. To address these problems comprehensively, we have implemented two core revisions: First, we **systematically evaluated and updated all figures in the "Results" section**, identified those with poor readability or ineffective data presentation, and replaced them with more intuitive, information-rich versions—these updated figures prioritize clear labeling (e.g., axes, legends, sample groups), logical data organization (e.g., grouped bar charts for comparative data, line charts for trend analysis), and direct alignment with the key points they aim to illustrate. Second, we have thoroughly reviewed and **revised the descriptive text for all figures in the section**: we supplemented key details (e.g., experimental context, sample sizes, statistical indicators) for each figure, clarified the meaning of core data points or trends, and explicitly linked the figure content to the corresponding research conclusions—ensuring that readers can easily follow how the visual data supports our arguments.

All revised figures and tables in the "Results" section, along with their updated descriptive text, can be found in our revised version. We reviewed each figure and its description multiple times to ensure consistency, clarity, and validity of the of presentation.

Reviewer #2

General comment

- RC:** *This manuscript presents a novel multimodal endoscopic sensing platform, the T-scope, which integrates visual, tactile, and thermal sensing into a single functioning device. The work addresses a longstanding technological limitation of conventional endoscopes, namely their inability to provide tactile and temperature perception alongside visual imaging. The authors report a technically innovative solution using tellurium thermocouples and viscoelastic silicone encapsulation, further enhanced by optimized heterointerfaces and deep neural network algorithms. As a result, the device is capable of allowing high-quality imaging, precise thermal mapping, and force feedback at microscale resolution. Moreover, the T-scope is compatible with standard and commercial endoscopes, and clinical verification has been performed through in-vivo and in vitro experiments. The manuscript is well-structured and technically rigorous; however, a few minor revisions would further enhance its overall clarity and readability.*
- AR:** Thank you sincerely for your positive recognition of our work—we greatly appreciate your acknowledgment of the T-scope’s novelty as a multimodal endoscopic sensing platform, its technical innovations (e.g., tellurium thermocouples, viscoelastic silicone encapsulation), and its value in addressing conventional endoscopes’ limitations. **We fully agree that minor revisions will enhance the manuscript’s clarity and readability, and thus have conducted a comprehensive revision of the entire text, focusing on three key aspects: enriching the tellurium thermocouples pattern design, optimizing the AI endoscopic tactile perception algorithm, and performing real animal endoscopic palpation experiments.** All these revisions, along with overall polishing for clarity, can be found in our revision, and we hope they further strengthen the manuscript’s quality.

Comment #1

- RC:** *The introduction would benefit from a clearer comparison with existing benchmarks and state-of-the-art approaches. In particular, an explicit discussion of tellurium relative to alternative materials would strengthen the rationale for selecting this material.*
- AR:** Thanks to the reviewers for their valuable suggestions. Mainstream temperature detectors are broadly categorized into contact and non-contact types. While infrared temperature sensors could circumvent direct contact with the lesion, their own accuracy has been the subject of criticism. In comparison, the development and application of contact temperature sensors have reached a relatively advanced stage of maturity. Among these, the Pt, Cu, and other precious metals represent the thermistor-type temperature sensors, exhibiting an exceptionally high signal stability. However, the signal itself is relatively less variable, and the high-precision means of detection are more complex. Furthermore, achieving high-precision signal detection requires complex bridge circuit calibration, which necessitates multiple lead wires—making implementation on flexible micro-probes impractical. Additionally, ensuring spatial measurement accuracy and minimizing external circuit variations demands constructing high-resistance sensitive units within a 1 mm diameter circle, with multiple wires extending outward. This necessitates the introduction of micro/nano fabrication techniques, significantly increasing process complexity. In contrast, thermocouple sensors inherently bind temperature to contact depth due to their mechanism. Consequently, thermocouple sensors eliminate the need for intricate patterned wiring designs and can be customized for visual-tactile sensing requirements, aligning with our needs.

Existing research indicates that tellurium and its compounds exhibit outstanding thermoelectric properties. Notably, recent studies demonstrate that tellurium nanomaterials have extremely low thermal conductivity

(facilitating pronounced temperature gradients) and exceptional Seebeck coefficients (yielding higher signal outputs for the same temperature difference), making them highly suitable for thermoelectric sensing unit fabrication. This is the rationale behind selecting Te thin films for device construction and optimization in this work. For a full explanation of the system and material selection, we have thoroughly revised the introduction section and provided tables in the supporting information for material comparisons along with relevant explanations (Supplementary Note S4, Table S6).

Revision:

INTRODUCTION: "Among the mainstream temperature measurement methods, non-contact temperature sensing represented by infrared temperature sensors is favored because it avoids contact damage, but the measurement accuracy is limited¹⁵⁻¹⁷. In contrast, contact-based temperature sensors are relatively mature in development and application, primarily categorized into thermistors and thermocouples. Thermistor made from precious metals like Pt and Cu offer exceptional signal stability¹⁸⁻²⁰. However, their minimal signal variation and complex calibration circuits make them difficult to integrate onto flexible micro-probes for visual-tactile sensing, rendering them more suitable for temperature drift calibration in micro-nano electronic devices²¹⁻²⁴. Thermocouples based on the Seebeck effect show a strong correlation between their signal and contact position and have excellent spatial resolution²⁵⁻²⁷. Leveraging the high band degeneracy and unique lattice arrangements of tellurium materials, a series of outstanding thermoelectric materials have been developed around tellurium and its compounds²⁸⁻³². However, due to thermoelectric mechanisms, thermocouple systems represented by tellurium-based compounds typically require larger longitudinal or lateral dimensions to achieve greater temperature differentials. This inevitably impacts signal acquisition and reconstruction in tactile sensing. Therefore, the construction of 3-dimensional, small-size, high-sensitivity temperature sensor devices remains an extremely significant challenge."

Supplementary Note S4: Thermoelectric effect. Temperature sensors are primarily categorized into non-contact and contact types. Non-contact sensors are susceptible to external interference and significant detection errors due to variations in the emissivity of the target object. Therefore, contact-type temperature sensors are preferred.

Contact-type temperature sensors mainly include thermistors and thermocouples. Thermistors primarily utilize precious metals and semiconductors as sensitive materials, measuring temperature by detecting changes in their internal resistance under thermal influence. These sensors typically exhibit minimal internal resistance changes, resulting in relatively small variations in the generated electrical signal. Furthermore, achieving high-precision signal detection requires complex bridge circuit calibration, which necessitates multiple lead wires—making implementation on flexible micro-probes impractical. Additionally, ensuring spatial measurement accuracy and minimizing external circuit variations demands constructing high-resistance sensitive units within a 1 mm diameter circle, with multiple wires extending outward. This necessitates the introduction of micro/nano fabrication techniques, significantly increasing process complexity. In contrast, thermocouple sensors do not require complex wiring design and can be structurally customized according to visual and tactile perception requirements to meet our needs.

Theoretical and experimental research on thermocouple sensing materials is relatively mature. Existing research indicates that tellurium and its compounds exhibit outstanding thermoelectric properties (Supplementary Table S6). Notably, recent studies demonstrate that tellurium nanomaterials have extremely low thermal conductivity (facilitating pronounced temperature gradients) and exceptional Seebeck coefficients (yielding higher signal outputs for the same temperature difference), making them highly suitable for thermoelectric sensing unit fabrication. This is the rationale behind selecting Te thin films for device construction and optimization in this work.

SI References: " 7. Zhou, M., Su, H., Pei, J. et al. Ultrahigh thermoelectricity obtained in classical BiSbTe

Supplementary Table S6. Thermal Electric Material Comparison

Material	Structure (Horizontal/ Vertical)	Sensitivity ($\mu\text{V/K}$)	Process temperature ($^{\circ}\text{C}$)	Dimensions	References
BiSbTe	V	200	610	2.6 mm	Ref.7
$\text{Bi}_2\text{Te}_3/\text{Bi}_{0.5}\text{Sb}_{1.5}\text{Te}_3$	H	354	250	5 μm	Ref.8
p-BiSbTe	V	240	600	10-12 mm	Ref.9
$\text{Bi}_{0.5}\text{Sb}_{1.5}\text{Te}_3$	V	240	410	12 mm	Ref.10
$\text{Bi}_2\text{Te}_{2.7}\text{Se}_{0.3}$	/	327	650	/	Ref.11
Bulk Te	V	400	550	1.5 mm	Ref.12
Te flake	H		180	5 μm	Ref.13
Te flake	H	413	180	50 μm	Ref.14
Te	V/H	647	Ambient temperature	~ 100 nm	This work

alloy processed under super-gravity. *Nat Commun* 16, 7645 (2025).

8. Gong, T. et al. High-Performance planar thin-Film thermoelectric cooler based on sputtered nanocrystalline $\text{Bi}_2\text{Te}_3/\text{Bi}_{0.5}\text{Sb}_{1.5}\text{Te}_3$ thin films for On-Chip cooling. *ACS Appl. Mater. Interfaces*. 17 (11), 17008–17017 (2025).

9. Zheng, G. et al. High thermoelectric performance of p-BiSbTe compounds prepared by ultra-fast thermally induced reaction. *Energ. Environ. Sci.* 10, 2638–2652 (2017).

10. Hao, F. et al. High efficiency Bi_2Te_3 -based materials and devices for thermoelectric power generation between 100 and 300 C. *Energ. Environ. Sci.* 9, 3120–3127 (2016).

11. Kim, J. H. et al. Possible Rashba band splitting and thermoelectric properties in CuI-doped $\text{Bi}_2\text{Te}_{2.7}\text{Se}_{0.3}$ bulk crystals. *J. Alloy Compd.* 806, 636–642 (2019).

12. Lin, S., Li, W., Chen, Z. et al. Tellurium as a high-performance elemental thermoelectric. *Nat Commun* 7, 10287 (2016).

13. Wu, X., Tao, Q., Li, D. et al. Unprecedentedly low thermal conductivity of unique tellurium nanoribbons. *Nano Res.* 14, 4725–4731 (2021).

14. Qiu, G. et al. Thermoelectric Performance of 2D Tellurium with Accumulation Contacts. *Nano Lett.* 19, 1955–1962 (2019)."

Comment #2

RC: *Further discussion on the stability, biocompatibility, and cost estimation of the proposed system would be valuable to assess its translational potential.*

AR: Thank you for your professional and insightful suggestion. We have significantly expanded the **Discussion section** of the revised manuscript—with a focus on advancing the translational application of the T-scope system—to address these key aspects in detail, and also supplemented relevant content on subsequent translational challenges.

Revision:

DISCUSSION: "We specifically discuss its performance and feasibility in three core dimensions: stability, biocompatibility, and cost. **Stability:** Our experimental results show that a single probe can reliably complete over 10k+ data acquisition cycles and support more than 20+ full animal experiments while maintaining operational functionality; to enhance structural stability, we design an integrated framework with both support and connection functions, and fully encapsulate the entire structure with Ecoflex silicone. We also explicitly note that the probe currently lacks an additional overload-resistant structure and may be damaged under accidental force exceeding 5 N, but it is easily replaceable, and its multi-layer elastomer encapsulation ensures no components will remain in the body, effectively mitigating potential risks for practical use. **Biocompatibility:** The core materials of the T-scope system include Ecoflex elastomer and tellurium (Te): Ecoflex has mature biocompatibility verification data to confirm its safety for biomedical scenarios; As for Te, it is tightly encapsulated by two inner rubber layers and an additional outer silicone coating. This multi-layer protection completely prevents Te from coming into direct contact with human tissue, further eliminating biocompatibility issues for translational use. **Cost estimation:** We provide a clear breakdown: the material cost for manufacturing a single probe is approximately 5 USD, a low-cost advantage that strongly supports the system's potential for large-scale practical application in clinical settings."

Comment #3

RC: *A more detailed description of the experimental setup for temperature measurements would enhance transparency and reproducibility. Have the authors employed a thermocouple as a reference instrument to acquire the temperature?*

AR: Thank you for your constructive suggestions. As noted, this work employs a Peltier as the heat source during sensor unit testing and utilizes a thermocouple to calibrate temperatures at different locations (Fig. 2k). To minimize calibration errors from varying contact positions and configurations, the thermocouple probe was fixed to the Peltier, with the thermosensitive unit covering the probe to simulate real-world conditions as closely as possible. Different bias voltages were applied to the Peltier to achieve varying temperatures. After waiting for 3 minutes, the corresponding temperature value was recorded as the contact temperature. Following contact temperature acquisition, the thermosensitive unit was positioned between the thermocouple probe and the Peltier. Similarly, different bias voltages were applied to the Peltier device. After waiting 3 minutes, the corresponding temperature values were recorded as the upper-side temperatures. Based on this, the temperature gradient could also be obtained. Considering this subsystem is primarily for temperature sensing, subsequent integrated device calibrations only collect temperatures at the contact end.

To facilitate controllable temperature calibration, a commercially available temperature-controlled hot plate was subsequently used as the heat source to calibrate the temperature sensing subsystem within the integrated device. To evaluate the stability of the thermostat, we set the thermostat to a specific temperature and maintained it at that temperature for 3 minutes before measuring the surface temperature using a thermocouple.

The measured deviation of 0.1°C demonstrated good temperature control characteristics and was suitable for subsequent temperature calibration. To avoid misunderstandings, we have provided a more detailed description of the temperature measurement in the revised "Methods" section.

Revision: Electrical Characterization of Te-based thermoelectric subsystem: We use a semiconductor analyzer (Keithley B1500A) to record the thermoelectric potential output of the sensing unit. We use a Peltier element as a single-side heat source, acquiring different temperature values through voltage control and calibrating with a thermocouple (UNI-T, UT325). To maintain the target contact temperature, we connect a thermocouple probe to the Peltier element and placed a thermistor above the probe (Supplementary Note S4, Table S6). After applying a bias voltage to the Peltier element for 3 minutes, the corresponding temperature is recorded as the contact temperature. The opposite-side temperature is obtained by placing a Te-based thermistor between the thermocouple probe and the Peltier element. After applying a bias voltage to the Peltier element for 3 minutes, the corresponding temperature is recorded as the opposite-side temperature. To facilitate temperature calibration and enhance controllability, a commercially available temperature-controlled hot plate is subsequently used as the heat source to calibrate the temperature sensing subsystem within the integrated device.

Comment #4

RC: *The readability of Figures 2 and 3 is limited; incorporating a legend and clearer annotations would substantially improve their interpretability.*

AR: Thank you for your professional comments. We have thoroughly revised both figures in the revised manuscript: First, we added comprehensive legends to Fig. 2 and 3, which explicitly explain the meaning of each curve, symbol, color, and component in the figures—ensuring readers can quickly associate visual elements with key experimental data or results. Second, we optimized the annotations by refining the wording for clarity, adjusting the placement to avoid overlap with figure content, and enhancing the font size for better readability.

Specifically, we revised Figures 2 and 3, and supplemented the previously missing sub-figures in Figures 2c 2e, and 3c.

Revision:

RESULTS (2.2): "This unique crystal structure exhibits significant lattice scattering, effectively suppressing lattice heat conduction and promoting the establishment of a significant temperature gradient (**Fig. 2c**). This enables efficient thermoelectric conversion and reduces the trade-off between thickness and responsivity. The fabrication process flow comprises the following key steps (Fig. 2d, Further details in the Methods section and Supplementary Fig. S3): Initial surface cleaning and PDMS composite treatment of 10 μm PET substrates; Magnetron sputtering deposition of gold electrodes on half-area substrates followed by full-substrate tellurium thin film deposition; Laser direct writing processing of AuTe-Te interfaces into 1 mm-diameter temperature-sensitive unit (**Fig. 2e**). When placed on a 42 °C hot plate, the temperature-sensitive unit demonstrates clear temperature field distribution characteristics on its surfaces, confirming temperature detection capabilities."

Fig. 2 | Mechanism and evaluation of the thermoelectric subsystem. **a** The structure of the Te-based thermoelectric subsystem. **b** The lattice structure diagram of Te. **c** Thermal gradient under 42 °C thermal loading (Upper: Macroscopic observation; Lower: Infrared thermographic mapping). **d** Fabrication process flow for polycrystalline Te temperature detector (Magnetron sputtering thin-film deposition combined with laser-assisted micro-patterning). **e** A shot of the finished Te-based thermoelectric sensor. **f** Optical image of magnetron-sputtered ITO films deposited on flexible PET substrates, demonstrating high optical clarity and visible-light transmittance. **g** Computational simulation illustrating the working principle of thermoelectric sensing elements, incorporating an elastomeric contact layer at the top (to simulate the thermoelectric subsystem immobilized on the tactile interface) and a thermally regulated boundary at the bottom (to simulate different contact temperatures), with colormap visualization of the thermal distribution. The figure utilizes a contact temperature of 30 °C and an ambient temperature of 25 °C. **h** The seebeck-induced voltage mapping within the functional layers, quantitatively correlated with interfacial temperature modulation at the active contact region, 20-40 °C. **i** Distribution of thermal and electrical potentials under 30 °C contact temperature along the upper end of the subsystem. **j** Time-dependent voltage response characteristics of the device under varying thermal differentials, defined as the difference between the contact temperature and the ambient temperature. **k** Quantitative dependence of generated electrical potential on applied temperature.

RESULTS (2.4): "Experimental results demonstrate that the system achieves excellent performance under varying contact magnitudes and directions (Supplementary Table S3 and S4). The average errors of the contact tangential and normal force components are 0.02 N and 0.6 N, respectively. Error distribution analysis reveals consistent prediction accuracy across different force magnitudes, with no error escalation observed when contact forces increase (**Fig 3c(i)**). Directional error analysis (Supplementary Fig. S9) indicates comparable performance across different orientations, though slightly higher errors occur in the central and downward directions. This phenomenon may be attributed to the contact habits of operators. Further validation through randomized finger contact trials (**Fig. 3c(ii)**) shows stable 3D force estimation with consistently low errors, confirming the generalization capability of the network."

Fig. 3 | AI-driven visual-tactile deep neural networks. **a (i)** The tactile dataset collection setup. **(ii)** Two different types of thermoelectric imprints of the T-scope probe. **(iii)** 18 soft materials used to collect 3D force-tactile image samples, with hardness covering various types of human tissues from 0 to 100 KPa. **b** The EndoForce network. **c (i)** Estimated force error distribution (contact force magnitude). **(ii)** Comparison of predicted force and true force when a person's finger randomly touches the probe. **d (i)** The coating-free T-scope sensor to collect visual inpainting dataset. **(ii)** The ProPainter network. **e (i)** Restored PSNR distribution (coating location and coating size). **(ii)** Visualization of visual inpainting results under various in-vivo scenarios.

Comment #5

RC: *The authors are encouraged to discuss whether, in addition to visual feedback, the integration of alternative modalities—such as advanced haptic feedback systems—could further strengthen the closed-loop performance of the instrument. Including such considerations in the discussion section would highlight opportunities for future development.*

AR: Thank you for this insightful suggestion. We have supplemented the **Discussion section** of the revised manuscript with a dedicated analysis of this aspect, aiming to highlight opportunities for future research and development while enriching the comprehensiveness of our performance evaluation.

Revision: DISCUSSION: "Future development will focus on developing an endoscopy system with tactile feedback, enabling operators to directly judge the hardness, temperature, and other conditions of the contacted tissue in real time through tactile force feedback from their hands, thereby improving diagnostic efficiency."

Reviewer #3

General comment

RC: *This work proposes a tellurium-based thermoelectric visual-tactile endoscope that integrates imaging, temperature sensing, and force feedback into a single platform. While the system is promising for multifunctional endoscopic applications, some questions remain to be addressed.*

AR: Thank you sincerely for your positive evaluation of our work—we greatly appreciate your recognition of the tellurium-based thermoelectric visual-tactile endoscope’s potential for multifunctional endoscopic applications, as well as your reminder of remaining questions to address.

We have implemented targeted optimizations in the revised manuscript, including refining the system’s structural design, upgrading the supporting AI algorithm, and correcting animal experiment details by explicitly presenting the real in-vivo palpation experiments we conducted. These adjustments aim to better address existing questions and further enhance the system’s performance and reliability.

Comment #1

RC: *The definitions of ‘contact beginning,’ ‘stable contact,’ ‘fast contact,’ and ‘slow contact’ are ambiguous. This makes it difficult to interpret the results of the in vitro contact experiment. Please clearly define these terms, for example, by specifying them along the time axis, to improve the clarity of the analysis.*

AR: Thank you for your professional comment. We have thoroughly revised the relevant content in the manuscript: We completely updated **Figure 4** by integrating a detailed time axis, on which each contact state is clearly marked with specific time nodes and corresponding force/touch signal changes—for example. Meanwhile, we revised the corresponding text descriptions in the Results section to align with the time-axis-based definitions, ensuring consistency between visual presentation and written explanation and significantly improving the clarity of the experiment analysis.

Revision:

To verify the practical efficacy of the T-scope system in clinical endoscopic environments, we conduct ex vivo experiments using full-scale biosimulation phantoms. During the experiments, the T-scope sensing device is mounted at the distal end of a standard endoscopic system. Operators perform multiple contact detection tests by manually manipulating the endoscope within the simulated models of the human bronchus, stomach, and large intestine (Fig. 4a(i), Fig. 4b(i), Fig. 4c(i)). This study specifically records the detection data from the sensor’s interaction with physiological and pathological tissues in these anatomical structures, and conducts comparative analyses on the dual-modal sensing capabilities (three-dimensional contact force and vision) of the T-scope system when it contacts normal tissues and simulated tumor tissues.

Biomechanical test data from the bronchial and gastric models (Fig. 4a(ii), b(ii)) shows that due to the higher biomechanical stiffness of tumor tissue, the fluctuation of its normal force value is more significant than that of healthy tissue when maintaining a stable contact state. Visual inpainting results (Fig. 4a(iii), b(iii)) indicate that the inpainting algorithm in this study can effectively eliminate the occlusion of normal and abnormal tissues caused by thermoelectric devices, enabling the normal visual observation function of the endoscope. In addition, in the human intestinal model (Fig. 4c(i)), we also conduct fast and slow contact tests on normal tissues: Fig. 4c(ii) and (iii) respectively show the comparison of normal force and tangential force sensed by the T-scope probe during fast contact and slow contact. Experimental results show that the increase rate of normal force (-1.4 vs -0.17) and the vibration amplitude of lateral force (-58 mN vs 21 mN) during fast contact are significantly greater than those during slow contact, which indirectly proves the sensitivity and

accuracy of the 3D force sensing deep neural network in this study. For more detailed three-dimensional force sensing data, please refer to Supplementary Fig. S13 and S14, Movie S3, S4, and S5.

Fig. 4 | In vitro contact experiment simulating human organ model. **a (i)** Simulated bronchial tissue palpation is performed using a manually operated endoscope system equipped with a T-scope probe. **(ii)** Normal force sequences are measured in normal tissue and simulated tumor tissue at the same contact depth while the operator manually feeds at similar speeds. **(iii)** Restoration of visual observation during contact (partial) in a (ii). **b (i)** Simulated stomach tissue palpation. **(ii)** Normal force sequences are measured in normal tissue and simulated tumor tissue at the same contact depth while stable contact. **(iii)** Restoration of visual observation during contact (partial) in b(ii). **c (i)** Simulated intestine tissue palpation. **(ii)** The z-axis force series in normal tissue and simulated tumor tissue are measured at the same contact depth while the operator manually feeds at fast and slow speeds. **(iii)** The x-axis force series in normal tissue and simulated tumor tissue are measured at the same contact depth while the operator manually feeds at fast and slow speeds.

Comment #2

RC: *Some figures included in the main manuscript are not cited in the text. Please check all figure references to ensure every figure is properly called out and discussed.*

AR: Thank you for your careful comment. We have conducted a comprehensive check of all figure references in the revised manuscript, with a focused effort to address issues in Figures 2, 3, and 4: We added targeted text descriptions to explicitly cite and discuss these three figures—explaining the core data, trends, and implications presented in each—and corrected any inconsistencies between the figure content and the

corresponding written analysis. This ensures every figure is properly called out, and its role in supporting the manuscript's conclusions is clearly articulated.

Specifically, we revised Figures 2 and 3, and supplemented the previously missing sub-figures in Figures 2c, 2e, 3c, 4e, 4f, and 4g. Please note that Figures 4e, 4f, and 4g in our original manuscript have been modified to Figures 4b(ii), 4b(iii), and 4c(i) in the revised manuscript.

RESULTS (2.2): "This unique crystal structure exhibits significant lattice scattering, effectively suppressing lattice heat conduction and promoting the establishment of a significant temperature gradient (**Fig. 2c**). This enables efficient thermoelectric conversion and reduces the trade-off between thickness and responsivity. The fabrication process flow comprises the following key steps (**Fig. 2d**, Further details in the Methods section and Supplementary Fig. S3): Initial surface cleaning and PDMS composite treatment of 10 μm PET substrates; Magnetron sputtering deposition of gold electrodes on half-area substrates followed by full-substrate tellurium thin film deposition; Laser direct writing processing of AuTe-Te interfaces into 1 mm-diameter temperature-sensitive unit (**Fig. 2e**). When placed on a 42 °C hot plate, the temperature-sensitive unit demonstrates clear temperature field distribution characteristics on its surfaces, confirming temperature detection capabilities."

RESULTS (2.4): "Experimental results demonstrate that the system achieves excellent performance under varying contact magnitudes and directions (Supplementary Table S3 and S4). The average errors of the contact tangential and normal force components are 0.02 N and 0.6 N, respectively. Error distribution analysis reveals consistent prediction accuracy across different force magnitudes, with no error escalation observed when contact forces increase (**Fig 3c(i)**). Directional error analysis (Supplementary Fig. S9) indicates comparable performance across different orientations, though slightly higher errors occur in the central and downward directions. This phenomenon may be attributed to the contact habits of operators. Further validation through randomized finger contact trials (**Fig. 3c(ii)**) shows stable 3D force estimation with consistently low errors, confirming the generalization capability of the network."

RESULTS (2.5): "Biomechanical test data from the bronchial and gastric models (**Fig. 4a(ii), b(ii)**) shows that due to the higher biomechanical stiffness of tumor tissue, the fluctuation of its normal force value is more significant than that of healthy tissue when in contact and maintaining a stable state. Visual inpainting results (**Fig. 4a(iii), b(iii)**) indicate that the inpainting algorithm in this study can effectively eliminate the occlusion of normal and abnormal tissues caused by thermoelectric devices, enabling the normal visual observation function of the endoscope. In addition, in the human intestinal model (**Fig. 4c(i)**), we also conduct fast and slow contact tests on normal tissues: **Fig. 4c(ii)** and **(iii)** respectively show the comparison of normal force and tangential force sensed by the T-scope probe during fast contact and slow contact. Experimental results show that the increase rate of normal force (-1.4 vs -0.17) and the vibration amplitude of lateral force (-58 mN vs 21 mN) during fast contact are significantly greater than those during slow contact, which indirectly proves the sensitivity and accuracy of the 3D force sensing deep neural network in this study."

Comment #3

RC: *A significant number of figures contain a variety of typographical and display errors. For instance, in Figure 4i, the label "contact beginning" is improperly positioned and overlaps with other text. It is highly recommended that the authors perform a comprehensive and meticulous review of all figures, including the supplementary figures, to correct these issues. Ensuring the accuracy of all values, labels, and legends is crucial for the credibility of the presented data.*

AR: Thank you for your expert feedback. We conducted a thorough and detailed review of all major figures and supplementary figures in the revised manuscript. We corrected typographical errors (e.g., spelling errors in

labels or legends), repositioned text and labels to eliminate overlap (including relabeling "contact beginning" in Figure 4i), verified the accuracy of all numerical values (e.g., axis scales, data point annotations), and standardized the formatting of legends and symbols for consistency. This thorough revision ensures that each figure is visually accurate and clearly presented.

Author Response to Reviews of

Superelastic Tellurium Thermoelectric Coatings for Advanced

Trimodal Microsensing

Shaowei Cui, Linlin Li, Zi-Xin Huang, Yanzhe Yu, Mingxue Cai, Xiangyin Bao,
Chaofan Zhang, Tiandong Zhang, Long Cheng, Wenxuan Zhang, Zheng Lou, Shuo
Wang, Wen Gong, Chao-Feng Wu, Lili Wang, Yu Wang

RC: Reviewer Comment, AR: Author Response

Reviewer #1

Comment #1

RC: In Figure 3 c(i), the error range for z error appears significantly larger compared to x error and y error. Please explain why the error range is large only for the z-axis.

AR: We appreciate the reviewer's careful comments. **The z-axis has a larger absolute error because its measurement range (0-1 N) is much larger than the ranges of the x-axis and y-axis (-0.15-0.15 N).** For most axially mounted multi-dimensional force/tactile sensors, the normal force range is generally significantly greater than the tangential force range [1]. **This measurement range is sufficient for the needs of tactile endoscopy, as most in vivo detection forces are in the range of several hundred mN, for example, as mentioned by Guo et al., " The range of the applied force was 0-200 mN with an interval of 40 mN [2]."** Further literature review suggests that **full-size percentage error (%FS) is a more suitable metric for multi-axial force error analysis [3, 4].**

Therefore, we have updated the manuscript to report the error distribution in %FS. As shown in the revised figure below, **the z-axis error is comparable to those of the x and y axes.** In terms of data, the full-scale error percentages for the x, y, and z axes are 6.50 %FS, 5.09 %FS, and 5.96 %FS, respectively. Since the statistical deviations of **forces along the x and y axes are essentially the same**, we used a more academically consistent term in our revision, **uniformly representing them as shear force.** We apologize for the confusion caused by the previous presentation of absolute error.

Fig. 1 | The estimated force error distribution within different contact force ranges (updated from absolute error, N to full-scale percentage error, %FS).

References:

- [1] Lee H K, Chung J, Chang S I, et al. Normal and shear force measurement using a flexible polymer tactile sensor with embedded multiple capacitors[J]. Journal of Microelectromechanical Systems, 2008, 17(4): 934-942.
- [2] Guo J, Chen B, Zhang Z, et al. A novel flexible forceps with fbg-based force-sensing and feedback for endoscopic surgery[J]. IEEE Sensors Journal, 2025, 25(15): 28427 - 28434.
- [3] Aksoy B, Hao Y, Grasso G, et al. Shielded soft force sensors[J]. Nature Communications, 2022, 13(1): 4649.

[4] Zhang J, Yao H, Mo J, et al. Finger-inspired rigid-soft hybrid tactile sensor with superior sensitivity at high frequency[J]. Nature communications, 2022, 13(1): 5076.

Revision:

Fig. 3 | AI-driven visual-tactile deep neural networks. a (i) The tactile dataset collection setup. **(ii)** Two different types of thermoelectric imprints of the T-scope probe. **(iii)** 18 soft materials used to collect 3d force-tactile image samples, with hardness covering various types of human tissues from 0 to 100 KPa. **b** The EndoForce network. **c (i)** The estimated force error distribution (full-scale error percentage, %FS) within different contact force ranges. **(ii)** Comparison of predicted force and true force when a person's finger randomly touches the probe. **d (i)** The coating-free T-scope sensor to collect visual inpainting dataset. **(ii)** The ProPainter network. **e (i)** Restored PSNR distribution (coating location and coating size). **(ii)** Visualization of visual inpainting results under various in-

vivo scenarios.

ABSTRACT: "microstructured tactile/force feedback (range: 0-1 N (normal) and -0.15-0.15 N (shear), accuracy: ~6 %FS)."

RESULTS (2.4): "The average full-scale error percentage of the contact force is about 6 %FS, with the average errors of the tangential force and normal force components being 0.02 N (6.50%FS) and 0.06 N (5.96 %FS), respectively."

Comment #2

RC: The accuracy of the z-axis force seems most critical for measuring contact force. However, in Figure 3 c(i, ii) of this paper, the z error shows the largest deviation. Can a system with comparatively less accurate z error (compared to x error and y error) be used for a tactile endoscope? Please explain whether this error range is acceptable by comparing it with previous research.

AR: We thank the reviewer for this professional comment regarding the critical nature of Z-axis accuracy. **The resulting errors for the X, Y, and Z axes of our T-scope are 0.019 N (6.50 %FS), 0.015 N (5.09 %FS), and 0.060 N (5.96 %FS), respectively.**

We compare our system with previous studies on tactile diagnostic sensors for similar medical applications. Beccani et al. mentioned that " Calibration was 96% linear with a goodness-of-fit of 93% and **mean absolute error of 0.085 ± 0.096 N** [1]." Xu et al. stated that " Third, the tactile sensor exhibits small hysteresis effects in response to **normal force (Fig. 2I; hysteresis error: ~6%)** and shear force (fig. S28; hysteresis error: ~6%) [2]." Meanwhile, Kim et al. noted that " **The average relative errors of the forces correspond to 3.1%, 4.5%, and 8.9% of full-scale output F/T ranges,** respectively [3]."

References:

[1] Bandari N, Dargahi J, Packirisamy M. Miniaturized optical force sensor for minimally invasive surgery with learning-based nonlinear calibration[J]. IEEE Sensors Journal, 2019, 20(7): 3579-3592.

[2] Xu C, Wang Y, Zhang J, et al. Three-dimensional micro strain gauges as flexible, modular tactile sensors for versatile integration with micro-and macroelectronics[J]. Science Advances, 2024, 10(34): eadp6094.

[3] Kim U, Kim Y B, Seok D Y, et al. A surgical palpation probe with 6-axis force/torque sensing capability for minimally invasive surgery[J]. IEEE Transactions on Industrial Electronics, 2017, 65(3): 2755-2765.

In summary, the Z-axis force measurement accuracy of the proposed T-scope sensor is comparable to or even better than that of tactile/force sensors designed for similar medical applications in terms of absolute error (N) and full-scale relative error percentage (%) (0.060N (Ours) vs 0.085N, 5.96% (Ours) vs 6% and 8.9%). Specifically, our absolute error (0.061 N) is lower than the 0.085 N reported by Bandari et al. [1]. Regarding the relative full-scale error, our result (5.96%) falls well

within the acceptable range defined by previous studies, being comparable to the ~6% noted in [2] and notably lower than the 8.9% reported by Kim et al. [3]. Therefore, the current error range is acceptable for effective tactile endoscopy.

Comment #3

RC: *On lines 323-324, the paper states, "we conduct ex vivo experiments using full-scale biosimulation phantoms." It seems that actual biological tissues were not used, yet the term "ex vivo" was. Is the use of the term "ex vivo" appropriate here? Please explain this.*

AR: We appreciate the reviewer's correction. We agree that "ex vivo" is not the correct classification for the phantom-based experiments conducted in this study. Accordingly, we have deleted this term from the entire manuscript to avoid any confusion.

Revision:

RESULTS (2.5): "To verify the practical efficacy of the T-scope system in clinical endoscopic environments, we conduct visual-tactile diagnostic experiments using full-scale biosimulation phantoms."

Reviewer #2

Comment #1

RC: I have carefully reviewed the revised manuscript along with the authors' point-by-point responses. All major issues and suggestions raised in my initial review have been thoroughly and convincingly addressed. The revisions have notably strengthened the manuscript, particularly through the improved clarity and depth of the Introduction and Discussion sections. The scientific rigor, coherence, and overall presentation have been substantially enhanced. Furthermore, the figures have been made more readable, with clearer descriptions and labeling that eliminate previous ambiguities. I am fully satisfied with the current version and have no additional comments.

I therefore recommend the manuscript for acceptance in its present form.

AR: We strictly appreciate your positive evaluation and recommendation for acceptance. We also deeply value the insightful suggestions you provided throughout the review process. Taking your advice to heart, we plan to further enhance our work by integrating force/haptic feedback into the current tactile endoscopic system. We remain committed to advancing this technology and making continuous contributions to the community.

Reviewer #3

Comment #1

RC: The authors have addressed all the comments. I recommend the publication on Nature communications.

AR: We thank the reviewer for confirming that our revisions have addressed all concerns. We are honored by the recommendation for publication and grateful for the constructive guidance provided throughout the review process.

Author Response to Reviews of

Superelastic Tellurium Thermoelectric Coatings for Advanced

Trimodal Microsensing

Shaowei Cui, Linlin Li, Zi-Xin Huang, Yanzhe Yu, Mingxue Cai, Xiangyin Bao,
Chaofan Zhang, Tiandong Zhang, Long Cheng, Wenxuan Zhang, Zheng Lou, Shuo
Wang, Wen Gong, Chao-Feng Wu, Lili Wang, Yu Wang

RC: Reviewer Comment, AR: Author Response

Reviewer #1

Comment #1

RC: I have no further comments and recommend this manuscript for acceptance.

AR: We are truly delighted that our revised manuscript has met with your approval and wish to express our sincere gratitude for your insightful and constructive suggestions, which have been instrumental in substantially elevating the quality of our work. Thank you again for your valuable time and support.